# PKM2 functions as a histidine kinase to phosphorylate PGAM1 and increase glycolysis shunts in cancer

Yang Wang [ID][1,3], Hengyao Shu[1,3], Yanzhao Qu[1], Xin Jin[1], Jia Liu [ID][1], Wanting Peng [ID][1], Lihua Wang[1], Miao Hao[2], Mingjie Xia[1], Zhexuan Zhao[1], Kejian Dong[1], Yao Di[1], Miaomiao Tian[1], Fengqi Hao[1], Chaoyi Xia[1], Wenxia Zhang[1], Xueqing Ba [ID][1✉], Yunpeng Feng [ID][1✉] & Min Wei [ID][1✉]

## Abstract

**Phosphoglycerate mutase 1 (PGAM1) is a key node enzyme that diverts the metabolic reactions from glycolysis into its shunts to support macromolecule biosynthesis for rapid and sustainable cell proliferation. It is prevalent that PGAM1 activity is upregulated in various tumors; however, the underlying mechanism remains unclear. Here, we unveil that pyruvate kinase M2 (PKM2) moonlights as a histidine kinase in a phosphoenolpyruvate (PEP)-dependent manner to catalyze PGAM1 H11 phosphorylation, that is essential for PGAM1 activity. Moreover, monomeric and dimeric but not tetrameric PKM2 are efficient to phosphorylate and activate PGAM1. In response to epidermal growth factor signaling, Src-catalyzed PGAM1 Y119 phosphorylation is a prerequisite for PKM2 binding and the subsequent PGAM1 H11 phosphorylation, which constitutes a discrepancy between tumor and normal cells. A PGAM1-derived pY119-containing cell-permeable peptide or Y119 mutation disrupts the interaction of PGAM1 with PKM2 and PGAM1 H11 phosphorylation, dampening the glycolysis shunts and tumor growth. Together, these results identify a function of PKM2 as a histidine kinase, and illustrate the importance of enzyme crosstalk as a regulatory mode during metabolic reprogramming and tumorigenesis.**

**Keywords** Phosphoglycerate Mutase 1 (PGAM1); Pyruvate Kinase M2 (PKM2); Histidine Kinase; Phosphorylation; Glycolysis Shunts
**Subject Categories** Cancer; Metabolism

## Introduction

Metabolic reprogramming is critical for tumorigenesis, and is a hallmark of tumor cells (DeBerardinis and Chandel, 2016; Faubert et al, 2020; Hanahan and Weinberg, 2011; Martinez-Reyes and Chandel, 2021; Pavlova and Thompson, 2016; Vander Heiden and DeBerardinis, 2017; Zhu and Thompson, 2019). To date, the best-characterized phenomenon associated with metabolic reprogramming in tumor cells is the Warburg effect, which is also referred to aerobic glycolysis, a shift of the glucose metabolism type from oxidative phosphorylation to glycolysis even under normal oxygen concentrations (DeBerardinis and Chandel, 2020; Vander Heiden et al, 2009). Aerobic glycolysis fuels the biosynthesis of macromolecules by providing essential metabolic intermediates and thereby supports rapid and sustainable cell proliferation (DeBerardinis and Chandel, 2020; Vander Heiden et al, 2009). In most cancers, upregulated glycolysis directly reinforces glycolytic shunts, which in turn fuels DNA, lipid and protein synthesis (Pavlova and Thompson, 2016). For instance, glycolytic intermediates glucose-6-phosphate and 3-phosphoglycerate can respectively enter the pentose phosphate pathway (PPP) and serine synthesis pathway (SSP), which support macromolecule synthesis by producing metabolic intermediates, such as reduced nicotinamide adenine dinucleotide phosphate (NADPH), ribose-5-phosphate (R5P) and one-carbon units (Jiang et al, 2014; Locasale, 2013; Patra and Hay, 2014; Stincone et al, 2014; Yang and Vousden, 2016). NADPH is not only required for the generation of reduced glutathione, a well-known antioxidant that eliminates reactive oxygen species (ROS) but also involved in lipid and nucleotide synthesis (Fan et al, 2014). Both R5P and one-carbon units are the important building blocks for nucleotide biogenesis, which is obviously essential for DNA synthesis and therefore cell proliferation (Patra and Hay, 2014). Although accumulating evidence suggested an important role for the coordination between glycolysis and its shunts in tumor cells, the underlying mechanism still remains elusive.

Phosphoglycerate mutase 1 (PGAM1) is a unique glycolytic enzyme responsible for converting of 3-phosphoglycerate (3PG) to 2-phosphoglycerate (2PG) in glycolysis (Chaneton and Gottlieb, 2012a; Hitosugi et al, 2012; Ohba et al, 2020; Qu et al, 2017; Sun et al, 2018; Zhang et al, 2017). 3PG, the substrate of PGAM1, binds to and competitively inhibits 6-phosphogluconate dehydrogenase

[1]Key Laboratory of Molecular Epigenetics of the Ministry of Education (MOE), Northeast Normal University, 5268 Renmin Street, 130024 Changchun, Jilin, China. [2]Science Research Center, China-Japan Union Hospital of Jilin University, 126 Xiantai Street, 130033 Changchun, Jilin, China. [3]These contributed equally: Yang Wang, Hengyao Shu. ✉E-mail: baxq755@nenu.edu.cn; fengyp0108@nenu.edu.cn; weim750@nenu.edu.cn

(6PGD) in the PPP, whereas 2PG, the product of PGAM1, activates 3-phosphoglycerate dehydrogenase (PHGDH) in the SSP (Chaneton and Gottlieb, 2012a; Hitosugi et al, 2012). Interestingly, it is prevalent that PGAM1 activity is upregulated in various tumors (Hitosugi et al, 2012, 2013; Huang et al, 2019), which is parallel with the decrease in 3PG level and the increase in 2PG level. This results in the relief of 6PGD inhibition and the advancement of PHGDH, thereby enhances PPP and SSP simultaneously, which places PGAM1 at a critical node to coordinate glycolysis with its shunts (Chaneton and Gottlieb, 2012a; Hitosugi et al, 2012). Importantly, inhibition of the expression or activation of PGAM1 by shRNA or small molecule inhibitors suppresses glycolysis and anabolic biosynthesis, and reduces cell proliferation as well as tumor growth (Huang et al, 2019; Wen et al, 2019). Hence, targeting PGAM1, which may limit both glycolysis and its shunts, is considered as a killing-two-birds-with-one-stone strategy for cancer therapy (Chaneton and Gottlieb, 2012a; Hitosugi et al, 2012). It is already well acknowledged that the enzymatic activity of PGAM1 is primed through the phosphorylation of histidine 11 (H11) (Hitosugi et al, 2013; Liu et al, 2017). Increasing evidence of recent studies revealed that the level of phosphoenolpyruvate (PEP), a metabolite downstream of PGAM1 and generated from 2PG in glycolysis, is significantly upregulated in tumor cells as compared to that from adjacent normal tissues (Chinnaiyan et al, 2012). PEP-derived phosphate appears to be responsible for the phosphorylation of PGAM1 at H11 (Vander Heiden et al, 2010; Vander Heiden et al, 2011); yet, the molecular mechanism controlling PEP-dependent PGAM1 phosphorylation at H11 is still unknown.

Here we demonstrated that glycolytic enzyme PKM2 directly binds to PGAM1 and functions as a histidine kinase, transferring the phosphorus group from PEP to PGAM1 H11. In response to epidermal growth factor (EGF) stimulation, PGAM1 is phosphorylated by Src at tyrosine 119 (Y119), which facilitates PKM2 association. Upon Y119 mutation of PGAM1 or treatment of a pY119-containing cell-permeable peptide, PGAM1–PKM2 interaction is disrupted, resulting in impaired PGAM1 H11 phosphorylation and activity and compromised PPP and SSP. As a consequence, both DNA synthesis and lipid generation are reduced, leading to suppressed tumor growth.

## Results

### PKM2 is the histidine kinase that phosphorylates PGAM1 at H11

Due to the well-characterized link between PEP and PGAM1 H11 phosphorylation in tumor cells (Vander Heiden et al, 2010; Vander Heiden et al, 2011), PEP-consuming metabolic enzymes emerge into our consideration. We investigated the interactions of PGAM1 with two types of metabolic enzymes, pyruvate kinases (PKs) and enolase (ENOs), that are known to be able to use PEP as substrate (Dombrauckas et al, 2005; Reed et al, 1996). First, we examined the mRNA levels of PKs and ENOs, and found that *PKM* and *ENO1* were the predominantly transcribed genes of these two families in tumor cells, that encodes PKM1, PKM2 and ENO1 (Appendix Fig. S1A,B). Protein co-immunoprecipitation assay was conducted and the result showed that both PKM2 and ENO1, but not PKM1, were

able to interact with PGAM1 (Fig. 1A; Appendix Fig. S1C,D). The PEP-producing enzyme phosphoenolpyruvate carboxykinase (PCKs) was also investigated, and the result showed the major isoform PCK2 was not detectable in PGAM1-associated immunoprecipitate (Fig. 1A; Appendix Fig. S1E). To further validate the physical interaction of PGAM1 with PKM2 and ENO1, we performed pulldown (PD) and biolayer interferometry (BLI) assays with purified recombinant proteins and the result showed that only PKM2, but not ENO1, directly interacted with PGAM1 (Fig. 1B–E). PKM2–PGAM1 interaction was further confirmed in a panel of highly proliferating tumor cells, including A549, H1299, MCF-7, MDA-MB-231, AGS, and Jurkat, but hardly detectable in untransformed cells, including HBE-2, MCF-10A, GES-1, activated T and naive T cells (Figs. 1F and EV1A–C), suggesting a pan-cancer existence of the interaction between these two metabolic enzymes. Since human non-small cell lung cancer (NSCLC) was first documented for PEP-based PGAM1 phosphorylation (Vander Heiden et al, 2010), A549 cell line was chosen as a representative to investigate the significance and mechanism of PKM2–PGAM1 interaction in most experiments of the present study. Next, whether PKM2 was capable to catalyze PGAM1 phosphorylation was investigated. PGAM1 H11 phosphorylation was examined by using a specific antibody 3-phosphohistidine (3-pHis) that has been validated in the previous study (Fuhs et al, 2015). The result showed that H11N mutation completely abolished PGAM1 phosphorylation (Fig. EV1D), and the PGAM1 phosphorylation was correlated with its association with PKM2 (Figs. 1F and EV1A–C). When PKM2 was knocked down, PGAM1 H11 phosphorylation was markedly reduced. Ectopic expression of wild-type (WT) PKM2 fully rescued the reduction of PGAM1 H11 phosphorylation, however, the PKM2 K270M mutant that is defective for PEP binding (Bollenbach et al, 1999; Dombrauckas et al, 2005) failed to do so (Figs. 1G and EV1E). Taken together, data suggested an important role of PKM2 in the regulation of PGAM H11 phosphorylation.

To authenticate that PKM2 is able to phosphorylate PGAM1 by utilizing PEP as a phosphate donor, we established an in vitro kinase assay with purified PGAM1 and PKM2. We found that PKM2 indeed was able to phosphorylate PGAM1, however, PGAM1 histidine phosphorylation was abolished when H11 was mutated (Fig. 1H). As expected, PKM2-mediated PGAM1 H11 phosphorylation was abolished due to heating or acid treatment because histidine phosphorylation is sensitive to low pH and high temperature (Fuhs and Hunter, 2017; Hunter, 2022) (Fig. 1H). When PKM2 was replaced with ENO1 or PKM1, or PEP was substituted by ATP, PGAM1 H11 was failed to be phosphorylated (Fig. EV2A–C). In addition, an elevation in PGAM1 activity was observed in the presence of PKM2, which was in line with the increase in pyruvate production (Fig. 1I,J), supposed to be resulted from PEP-cleavage and phosphate donation to PGAM1. Notably, PKM2-upregulated PGAM1 activity and pyruvate production were impaired when H11 of PGAM1 was mutated (Fig. 1I,J). We also detected the interaction of PGAM1 with the nucleoside diphosphate kinase 1 and 2 (NME1 and NME2), that were reported to be able to phosphorylate histidines with ATP or GTP as phosphodonor in mammals (Fuhs and Hunter, 2017; Hunter, 2022), and neither of them could interact with PGAM1 or catalyze PGAM1 H11 phosphorylation, regardless of the presence of PEP, ATP or GTP (Fig. EV2D–H).

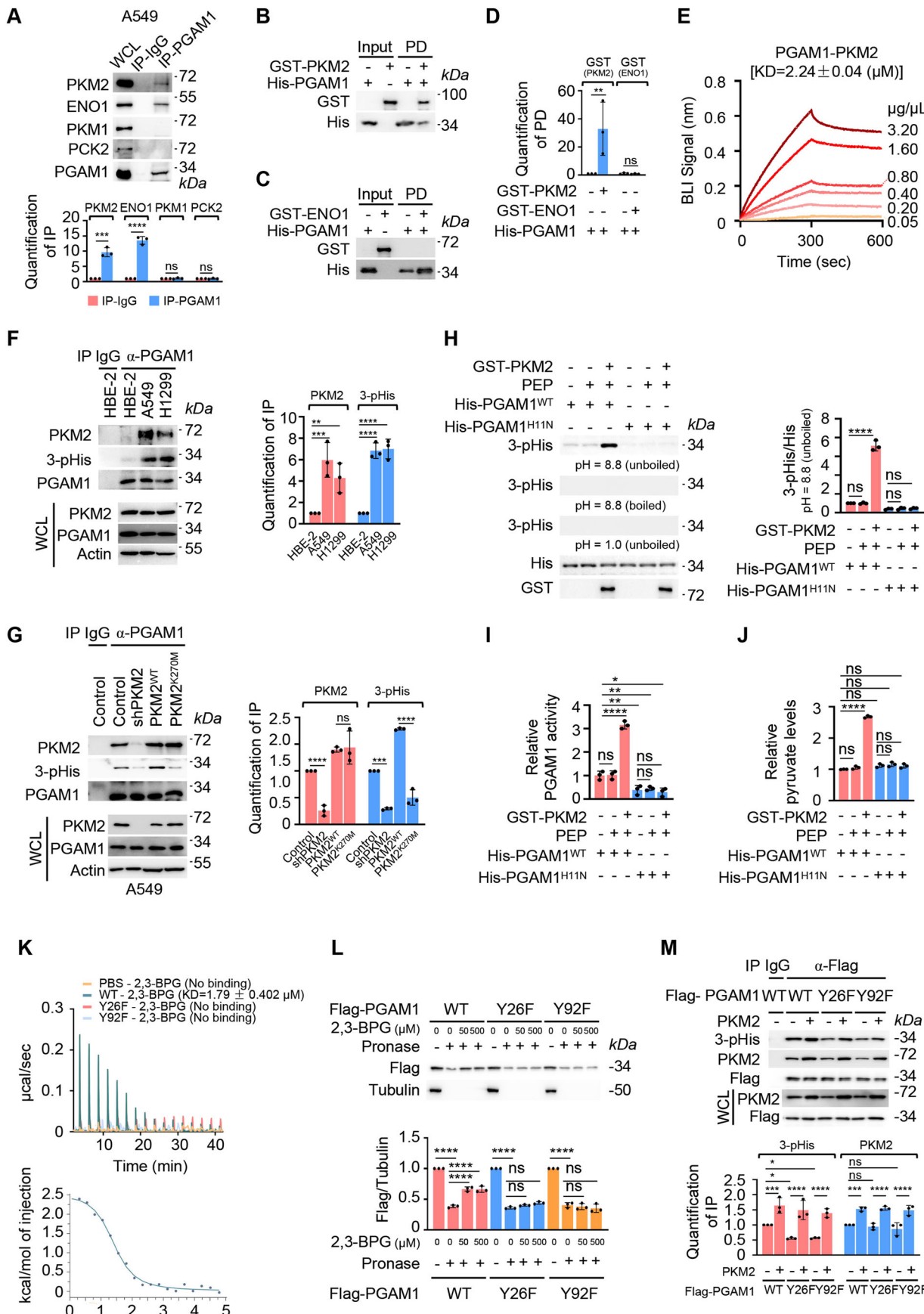

**Figure 1.  PKM2 is the histidine kinase that phosphorylates PGAM1 at H11.**

(A) PKM2 and ENO1 exist in the PGAM1-associated protein complex. Top: The whole-cell lysates (WCL) from A549 cells were prepared for immunoprecipitation (IP) and analyzed by western blot (WB). Bottom: Quantification of IP. Relative levels of PKM2, ENO1, PKM1, or PCK2 in IP were respectively normalized to that in WCL for each group. (B–D) PGAM1 directly interacts with PKM2 but not ENO1. Recombinant His-PGAM1 was immobilized on Ni-beads and then mixed with recombinant GST-PKM2 (B) or GST-ENO1 (C) prior to PD assay. The protein complex on the Ni-beads was eluted and analyzed by WB. (D) Quantification of PD in (B, C). Relative levels of GST-PKM2 or GST-ENO1 were normalized to that of His (PGAM1) for each group. (E) Confirmation of PGAM1 and PKM2 interaction by BLI assay. Recombinant His-PGAM1 protein was loaded onto Ni-sensor. The sensorgrams for the interaction of PGAM1 with GST-PKM2 as well as the associated KD values are shown as indicated. (F) PGAM1–PKM2 interaction and PGAM1 phosphorylation are more evident in tumor cells. Left: PGAM1-associated proteins were immunoprecipitated from the WCL of different tumor cells and analyzed by WB. Right: Quantification of IP. Relative levels of PKM2 or 3-pHis were normalized to that of PGAM1 for each group. (G) PKM2 K270M mutation reduces PGAM1 H11 phosphorylation. Left: PKM2 WT or K270M mutants were restored in endogenous PKM2-depleted A549 cells, PGAM1-associated proteins were immunoprecipitated and analyzed by WB. Right: Quantification of IP. Relative levels of PKM2 or 3-pHis were normalized to that of PGAM1 for each group. (H) PKM2 directly phosphorylates PGAM1 at H11. Top: In vitro kinase assays were carried out with recombinant GST-PKM2, His-PGAM1 as well as His-PGAM1 H11N mutant in the presence of PEP under normal (pH = 8.8, unboiled), heating (pH = 8.8, boiled) or acid condition (pH = 1, unboiled). PGAM1 H11 phosphorylation was detected by WB. Bottom: Quantification of WB. Relative levels of 3-pHis were normalized to that of His (PGAM1) for each group. (I, J) PKM2 upregulates PGAM1 activity. The PGAM1 activity (I) and pyruvate generation (J) from in vitro kinase assay in the normal condition as in (H) were detected by commercially available kits. (K) ITC binding measurements of 2,3-BPG with PGAM1. The recombinant His-PGAM1 proteins purified bacterially were injected into calorimetric cells before 2,3-BPG injection. Theoretical titration curves as well as associated KD values were displayed. (L) DARTS binding assay of 2,3-BPG with PGAM1. A549 cells were transfected with Flag-PGAM1 WT, Y26F or Y92F for 48 h. Top: The whole lysates were respectively incubated with the indicated doses of 2,3-BPG respectively for 30 min at room temperature followed by pronase digestion and WB analysis. Bottom: Quantification of WB. Relative levels of Flag were normalized to tubulin without pronase for each group. (M) PKM2 upregulates H11 phosphorylation of wild-type PGAM1 as well as Y26F and Y92F mutations. Top: A549 cells were transfected with Flag-PGAM1 WT, Y26F or Y92F for 48 h. Flag (PGAM1)-associated proteins were immunoprecipitated and analyzed by WB. Bottom: Quantification of IP. Relative levels of 3-pHis or PKM2 were normalized to that of Flag (PGAM1) for each group. Data information: for WB in (A–C, F–H, L, M), BLI in (E) and ITC in (K), one representative experiment out of three is shown. For IP in (A, F, G, M), IgG served as a negative control. Data were represented as mean ± SD ($n = 3$) with significance determined by one-way (F–J, L, M) or two-way (A, D) analysis of variance (ANOVA) test; ****$P < 0.0001$, ***$P < 0.001$, **$P < 0.01$, *$P < 0.05$. ns nonsignificant. Source data are available online for this figure.

2,3-Bisphosphoglycerate (2,3-BPG), an intermediate in PGAM1-catalyzed conversion of 3PG into 2PG, is responsible for phosphorylation of H11 by transferring its phosphate group at C3 position to H11. This raises the possibility that 2,3-BPG may contribute to PGAM1 H11 phosphorylation independent of PEP-derived phosphorus group donation since 2,3-BPG level is increased in PKM2-expressed cells as reported (Vander Heiden et al, 2010; Wiese and Hitosugi, 2018). To exclude the involvement of 2,3-BPG in the PKM2-catalyzed PGAM1 H11 phosphorylation, PGAM1 mutants, such as PGAM1$^{Y26F}$ and PGAM1$^{Y92F}$ were constructed to disrupt 2,3-BPG binding as described in the previous studies (Hitosugi et al, 2013; Wang et al, 2006; Wang et al, 2005). The isothermal titration calorimetry (ITC) measurement was employed, and the result verified that Y26F and Y92F mutations thoroughly inhibited the interaction between the recombinant PGAM1 and 2,3-BPG in vitro (Fig. 1K). In addition, the drug-affinity responsive target stability (DARTS) assay, that is used to detect the reduction in the protease susceptibility of the target proteins upon the small molecules binding, was conducted (Lomenick et al, 2009). A549 cell lysates containing WT PGAM1 or its mutants were incubated with increasing concentration of 2,3-BPG, then digested with pronase. The result showed that, 2,3-BPG was able to protect the WT PGAM1, but not that with Y26F or Y92F mutation from the degradation by pronase, implying a failure of 2,3-BPG binding with the mutants (Fig. 1L). Next, PKM2 were overexpressed along with WT PGAM1 or its mutants in A549 cells, and the PGAM1 H11 phosphorylation in the presence or absence of PKM2 was detected. Although compared with the WT PGAM1, H11 phosphorylation of Y26F and Y92F mutants significantly decreased, that was resulted from the disruption of 2,3-BPG binding with PGAM1 to provide phosphorus group, the interactions of the mutants with PKM2 were not impaired, and the overexpression of PKM2 could still significantly increase H11 phosphorylation of PGAM1$^{Y26F}$ and PGAM1$^{Y92F}$ mutants (Fig. 1M), suggesting that 2,3-BPG is not involved in PKM2-catalyzed

PGAM1 H11 phosphorylation. Taken together, combined data supported that PKM2 acts as a histidine kinase to phosphorylate PGAM1 at H11 in a PEP-dependent manner.

## De-tetramerization is favorable for PKM2 implementing PGAM1 H11 phosphorylation

PKM2 is the predominant PKs in tumor cells and acts as the rate-limiting enzyme controlling the final step of glycolysis (Christofk et al, 2008a). PKM2 either exists in the active tetramers that conventionally catalyzes PEP and ADP into pyruvate and ATP, or forms the less active dimers or inactive monomers to regulate other cellular processes to meet metabolic variations of the tumor cells (Dayton et al, 2016; Mazurek, 2011). To gain functional insights into the regulation of PGAM H11 phosphorylation by distinct PKM2 oligomers, we supplemented cells with PKM2 activator TEPP46 (Anastasiou et al, 2012) to stabilize PKM2 as tetramers, and found that both PKM2–PGAM1 interaction and PGAM1 H11 phosphorylation were blocked (Fig. 2A). It is quite reasonable that de-tetramerization gives rise to PKM2 accommodating an appropriate interface required for transferring phosphate from PEP to receptor proteins. In line with this observation, our mass spectrometry assay, where proteins associated with tetrameric PKM2 (with TEPP46) and de-tetrameric PKM2 (without TEPP46) were compared, revealed that PGAM1 was a highly scored protein associated with the de-tetramerized PKM2 (Fig. 2B). Furthermore, different PKM2 mutants, including K422R (tending to form tetramer), R399E (tending to form dimer) and Y105E (tending to form monomer) (Wang et al, 2015) were constructed and overexpressed in the cells lacking endogenous PKM2 (Fig. 2C). The result showed that the ratio of PKM2 tetramer significantly increased due to K422R substitution, which led to impaired PKM2–PGAM1 interaction and PGAM1 H11 phosphorylation; whereas, overexpression of PKM2 mutants bearing R399E or Y105E resulted in an enhanced PKM2–PGAM1 interaction along

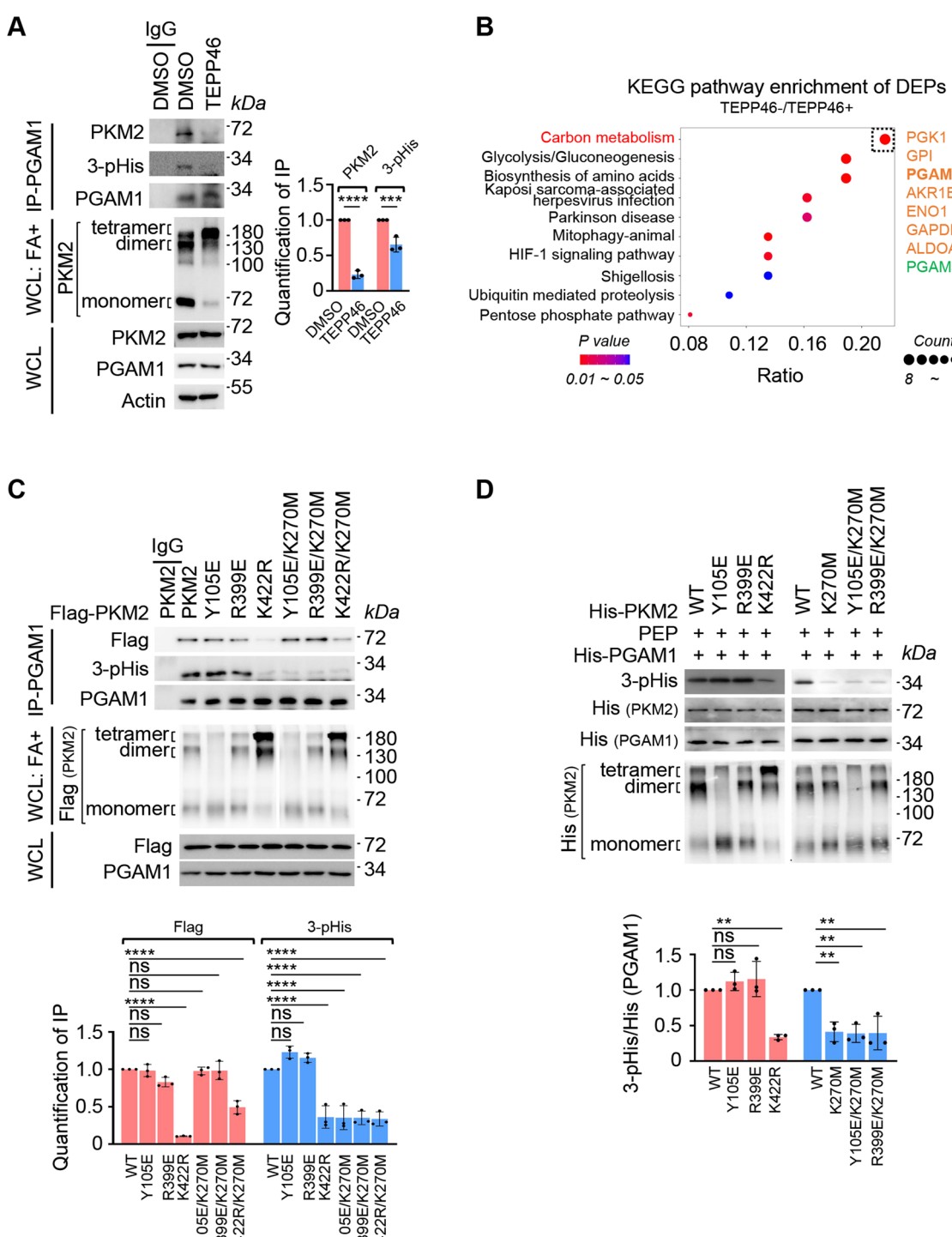

with PGAM1 H11 phosphorylation (Fig. 2C). In addition, the impacts of K422R, R399E and Y105E mutations on PGAM1 H11 phosphorylation was confirmed in vitro (Fig. 2D), and their influences on PGAM1 activity and pyruvate generation was validated as well (Appendix Fig. S2A,B). These observations suggested that PKM2 tetramerization blocks PGAM1 H11 phosphorylation, that is mediated by PKM2 dimer and monomer. Since K270 of PKM2 is required for PEP binding (Bollenbach et al, 1999;

Dombrauckas et al, 2005), we also interrogated K270 mutation along with those tested PKM2 mutants. Once K270 was mutated, PGAM1 H11 phosphorylation mediated by PKM2 R399E or Y105E disappeared, regardless of their association with PGAM1 (Fig. 2C,D). This further confirmed the PEP-dependent phosphorylation of PGAM1 at H11 by PKM2. The combined data suggested that de-tetramerization is favorable for PKM2 implementing PGAM1 H11 phosphorylation.

◄ **Figure 2. De-tetramerization is favorable for PKM2 implementing PGAM1 H11 phosphorylation.**

(A) Tetramerization of PKM2 reduces PGAM1 association and H11 phosphorylation. Left: A549 cells were treated with or without TEPP46 (50 µM) to stabilize PKM2 tetramer for 24 h. The whole-cell lysates were cross-linked with formaldehyde (FA) and the states of PKM2 oligomer were analyzed WB. PGAM1-associated proteins were immunoprecipitated and analyzed by WB. Right: Quantification of immunoprecipitation (IP). Relative levels of PKM2 and 3-pHis were normalized to that of PGAM1 for each group. (B) PKM2 interactome revealed by mass spectrometry (MS). A549 cells were treated with or without TEPP46 (50 µM) for 24 h. PKM2-associated proteins were immunoprecipitated and then analyzed by MS. The differentially expressed proteins (DEPs) between TEPP46 treatment or not (TEPP46-/TEPP46 +, twofold or more, $P < 0.05$, Student's $t$ test, $n = 3$ biological replicas) were subjected to Kyoto Encyclopedia of Genes and Genomes (KEGG) pathways enrichment assay and the result was displayed in bubbles plots. The bubble size indicates the enrichment ratio of DEPs in the pathway, and the bubble color indicates the $P$ value of significant enrichment. The DEPs in the top bubble were enriched in the carbon metabolism pathway and listed in descending order according to the fold change on the right side, in which the orange color indicated the significantly upregulated DEPs, and the green color indicated the significantly downregulated DEPs. (C) PKM2 K422R mutation blocks PGAM1 association and H11 phosphorylation. Top: Endogenous PKM2-depleted A549 cells were respectively transfected with Flag-PKM2 or its mutants including Y105E, R399E, K422R, Y105E/K270M, R399E/K270M and K422R/K270M for 48 h. The whole-cell lysates were cross-linked with FA and the states of PKM2 oligomer were analyzed by WB. PGAM1-associated proteins were immunoprecipitated and analyzed by WB. Bottom: Quantification of IP. Relative levels of Flag (PKM2) or 3-pHis were normalized to that of PGAM1 for each group. (D) PKM2 Y105E and R399E mutations are favorable for PGAM1 H11 phosphorylation. Top: In vitro kinase assay was carried out with recombinant His-PGAM1, His-PKM2, and its mutations, including Y105E, R399E, K422R, K270M, Y105E/K270M, R399E/K270M in the presence of PEP. PGAM1 H11 phosphorylation was detected by WB. His-PKM2 and its mutants were cross-linked with FA and the states of PKM2 oligomer were analyzed by WB. Bottom: Quantification of WB. Relative levels of 3-pHis were normalized to that of His (PGAM1) for each group. Data information: for WB in (A, C, D), one representative experiment out of three is shown. For IP in (A, C), IgG served as a negative control. Data were represented as mean ± SD ($n = 3$) with significance determined by one-way (C, D) two-way (A) ANOVA test; ****$P < 0.0001$, ***$P < 0.001$, **$P < 0.01$, ns nonsignificant. Source data are available online for this figure.

## PGAM1 Y119 phosphorylation is required for PKM2 recognition

Next, how the kinase recognizes its substrate arose as a question. Since PKM2 is a phosphotyrosine-binding protein (Christofk et al, 2008b), we asked whether PKM2–PGAM1 interaction relies on tyrosine phosphorylation of the latter. To this end, we used recombinant protein tyrosine phosphatase 1B (PTP1B) (Hitosugi et al, 2009) to eliminate tyrosine phosphorylation of PGAM1 and found that PKM2 association with PGAM1 was almost completely abolished, and the phosphorylation of PGAM1 at H11 was reduced significantly (Fig. 3A). These observations indicated that tyrosine phosphorylation in PGAM1 might be important for PKM2 association. To identify tyrosine phosphorylation sites in PGAM1, we performed mass spectrometry (MS) analysis with the proteins co-immunoprecipitated with PGAM1 from A549 cells. Although many tyrosine sites, including Y26, Y50, Y92, Y119, Y133 and Y142 (Fig. 3B; Appendix Fig. S3), were identified to be phosphorylated, only Y119 had a functional consequence on PGAM1–PKM2 interaction (Fig. 3C,D). Mutation of Y119 to unphosphorylatable phenylalanine (Y119F) markedly reduced PGAM1–PKM2 interaction (Fig. 3C); whereas when Y119 was replaced by glutamate (Y119E), a mutation widely used to mimic constitutive phosphorylation (Dionne et al, 2018; Kasahara et al, 2018; Zhou et al, 2020), this interaction was significantly improved (Fig. 3D). Importantly, both PKM2-mediated PGAM1 H11 phosphorylation and PGAM1 activity were reduced upon Y119 mutation (Fig. 3E,F). Intriguingly, Y119 in PGAM1 is conserved through mammals; whereas, the corresponding site of the homologs among the species other than mammals is mainly presented as phenylalanine that is structurally resembles with tyrosine (Fig. 3G). Implicit in this F-to-Y evolution in mammals, seems acquiring a layer of manipulation of enzymatic activity while leaving the conformation of the enzyme unaffected.

Y119 is structurally located at an external loop of PGAM1 (Hitosugi et al, 2013), thus Y119 phosphorylation might be beneficial to PKM2 binding and then catalyzing the phosphorylation of internal H11 of PGAM1 (Fig. EV3A). To directly detect the phosphorylation of PGAM1 at Y119, a customized antibody with specificity to recognize pY119 of PGAM1 was generated

(Fig. EV3B). By using this antibody, we observed that PGAM1 Y119 phosphorylation was much higher in tumor cells, including A549, H1299, MCF-7, and MDA-MB-231, than that in untransformed cells, including HBE-2 and MCF-10A (Figs. 3H and EV3C). Importantly, this is consistent with the observation that PGAM1–PKM2 interaction and PGAM1 H11 phosphorylation were enhanced in tumor cells (Figs. 1F and EV1A–C). In addition, we also noticed that PGAM1^WT and PGAM1^Y119F no longer had different impact on PGAM1 H11 phosphorylation if PKM2 was depleted (Fig. 3I), which further confirmed that PGAM1 Y119 phosphorylation is prerequisite for PKM2-inflicted PGAM1 H11 phosphorylation. Collectively, data suggest that PGAM1 Y119 phosphorylation is required for PKM2 recognition, and beneficial for tumor cells.

## EGF signaling accounts for PGAM1 Y119 phosphorylation

Evidently, PGAM1 Y119 phosphorylation constitutes the discrepancy between tumor cells and the normal proliferating ones, which is appealing an interpretation of the signaling mechanism that exists in tumor cells. Intriguingly, it is prevalent that EGF signaling is activated in many tumors such as lung cancers and breast cancers, and has shown to be important for aerobic glycolysis by regulating the glycolytic enzymes posttranslational modifications (Lv et al, 2013; Moscatello et al, 1995; Nicholson et al, 2001; Yang et al, 2012b). Herein, we further explored the role of EGF signaling in the regulation of PGAM1 by PKM2. In response to EGF signaling, increases in PGAM1 Y119 phosphorylation, the interaction of PKM2 with PGAM1, as well as PGAM1 H11 phosphorylation were observed (Figs. 4A,B and EV4A,B), however, which was completely abrogated once Y119 was mutated (Figs. 4B and EV4B). To identify the tyrosine kinase(s) responsible for EGF-induced PGAM1 Y119 phosphorylation, EGFR and the tyrosine kinases downstream of EGFR were individually inactivated by inhibitors including Afatinib (EGFR inhibitor), Ruxolitinib (JAK family inhibitor), Amuvatinib (c-Kit inhibitor), or Saracatinib (Src family inhibitor). The on-target effects of the inhibitors were validated (Liu et al, 2019) (Fig. 4C). Inhibition of EGFR by Afatinib was included as a positive control, and the abrogation of both

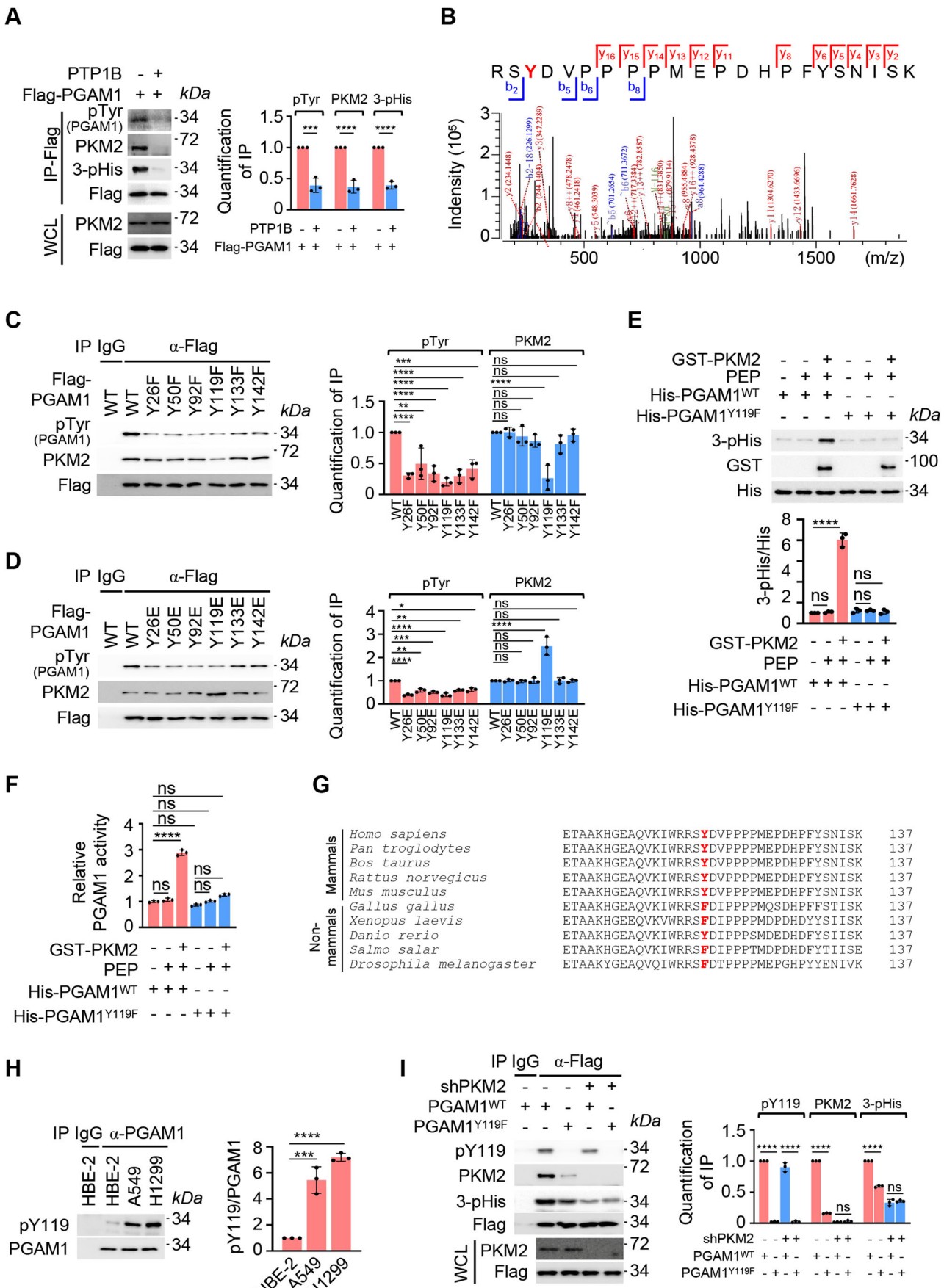

**Figure 3.   PGAM1 Y119 phosphorylation is required for PKM2 recognition.**

(A) Inhibition of PGAM1 tyrosine phosphorylation reduces PKM2 interaction. Left: A549 cells were transfected with Flag-PGAM1 for 48 h. Flag (PGAM1)-associated proteins were immunoprecipitated and immobilized on the agarose beads coated with flag antibody, and then respectively treated with or without protein tyrosine phosphatase 1B (PTP1B, 1 μg/μL) for 30 min. The protein complex was eluted and analyzed by WB. Right: Quantification of IP. Relative levels of pTyr, PKM2 or 3-pHis were normalized to that of Flag (PGAM1) for each group. (B) PGAM1 is phosphorylated at Y119 analyzed by MS. PGAM1 protein immunoprecipitated from A549 cells was trypsinized and then analyzed by MS. MS data were processed with the Byonic engine, and the peptide containing phosphorylated Y119 was identified. The y and b fragmentations were used to map the phosphorylation site to the Tyr indicated in red. (C, D) PGAM1 Y119 phosphorylation determines PKM2 association. Left: A549 cells were transfected with Flag-PGAM1 and its mutants for 48 h. Flag (PGAM1)-associated proteins were immunoprecipitated and analyzed by WB. Right: Quantification of IP. Relative levels of pTyr (PGAM1) or PKM2 were normalized to that of Flag (PGAM1) for each group. (E, F) PGAM1 Y119F mutation blocks PKM2-mediated PGAM1 H11 phosphorylation. In vitro kinase assay was carried out with recombinant GST-PKM2, His-PGAM1 and its Y119F mutant in the presence of PEP. (E, top) PGAM1 H11 phosphorylation level was detected by WB. (E, bottom) Quantification of WB. Relative levels of 3-pHis were normalized to that of His (PGAM1) for each group. (F) PGAM1 activity was analyzed by commercially available kits. (G) Tyrosine 119 is conserved through mammals. Amino acid sequences that cross Y119 of PGAM1 and its homologues from various species. Y119 or the corresponding site is highlighted in red. (H) PGAM1 Y119 phosphorylation is more evident in tumor cells. Left: From HBE-2, A549 and H1299 cells, PGAM1-associated proteins were immunoprecipitated and analyzed by WB. Right: Quantification of IP. Relative levels of pY119 were normalized to that of PGAM1 for each group. (I) PGAM1 WT and Y119F mutant no longer have impacts on PGAM1 H11 phosphorylation in the absence of PKM2. Left: Flag-PGAM1 WT or Y119F mutant was respectively transfected into normal A549 or PKM2-depleted A549 cells for 48 h. Flag (PGAM1)-associated proteins were immunoprecipitated and analyzed by WB. Right: Quantification of IP. Relative levels of pY119, PKM2 or 3-pHis were normalized to that of Flag (PGAM1) for each group. Data information: for WB in (A, C–E, H, I), one representative experiment out of three is shown. For IP in (C, D, H, I), IgG served as a negative control. Data were represented as mean ± SD ($n = 3$) with significance determined by one-way (C–E, F, H, I) or two-way (A) ANOVA test; ****$P < 0.0001$, ***$P < 0.001$, **$P < 0.01$, *$P < 0.05$, ns nonsignificant. Source data are available online for this figure.

PGAM1 Y119 phosphorylation and PKM2 binding was observed as expected (Fig. 4C). The addition of Saracatinib, but not Ruxolitinib or Amuvatinib (Liu et al, 2019), blocked the upregulation of PGAM1 Y119 phosphorylation as Afatinib did (Fig. 4C), indicating that Src, downstream of EGF signaling, accounts for PGAM1 Y119 phosphorylation. Besides EGF, other oncogenic growth factors, such as fibroblast growth factor (FGF) and platelet-derived growth factor (PDGF), could activate Src and widely function in various tumors (Cao, 2013; Ostman et al, 2005; Presta et al, 2017; Turner and Grose, 2010). Thus, the effects of different oncogenic growth factors on the induction of PGAM1 Y119 phosphorylation were further tested. FGF or PDGF stimulation promoted Y119 phosphorylation and PGAM1–PKM2 interaction, and the inhibition of FGF receptor (FGFR), PDGF receptor (PDGFR) or Src individually by Zoligratinib, Crenolanib or Saracatinib blocked the functions of FGF or PDGF (Fig. EV4C,D). In addition, overexpression of Src enhanced PGAM1 Y119 phosphorylation, PGAM1–PKM2 interaction as well as PGAM1 H11 phosphorylation in A549 cells that expressed wild-type PGAM1 but not PGAM1 Y119F (Fig. 4D), suggesting the commitment of Src in transmission of growth factors' signaling to induce PGAM1 Y119 phosphorylation. To further validate the function of Src, in vitro kinase assay was carried out and the result showed that only Src, but not EGFR, FGFR, PDGFR, directly phosphorylated PGAM1 at Y119 (Fig. 4E,F); however, Y119F mutation is unable to respond to Src-mediated-Y119 phosphorylation, thus failing to regulate H11 phosphorylation (Fig. 4F). Taken together, these data confirmed the function of Src in growth factors' signaling-induced PGAM1 Y119 phosphorylation, which in turn enhances PKM2–PGAM1 interaction as well as PGAM1 H11 phosphorylation.

## Glycolytic shunts and cell proliferation are compromised due to the mutation of PGAM1 Y119

To further corroborate the function of PGAM1 Y119 phosphorylation in tumor cell proliferation, endogenous PGAM1 was depleted from A549 cells, and PGAM1 activity was significantly

rescued by the restoration of PGAM1 WT but not Y119F mutant (Fig. 5A). As expected, PGAM1 Y119 mutation led to accumulation in upstream metabolic intermediates (e.g., glucose) and reduction in downstream intermediates (e.g., 2PG) (Fig. 5B). Importantly, [$^{13}C_6$]-glucose tracing experiment also revealed that metabolites in glycolysis shunts PPP and SSP, such as R5P and serine, were impaired upon PGAM1 Y119F mutation (Fig. 5B). Because of the critical role of PPP and SSP in nucleotides synthesis and NADPH production (Fan et al, 2014; Hitosugi et al, 2012; Jiang et al, 2014; Patra and Hay, 2014; Stincone et al, 2014), we further investigated the impact of PGAM1 Y119F mutation on these processes. The result showed that along with the impairment of glycolysis shunts PPP and SSP, metabolic intermediates in nucleotide synthesis, such as ATP and AMP, were also significantly decreased (Fig. 5B). This was consistent with the EdU incorporation assay that showed impaired DNA replication (Fig. 5C). The mitigation of NADPH production was also resulted from the mutation as expected (Fig. EV5A). NADPH is not only producing reduced forms of antioxidants, but also involved in lipid synthesis (Fan et al, 2014). Accordingly, an increase in ROS and a decrease in cellular lipid were also observed with Y119F mutation of PGAM1 (Figs. EV5B,C and 5D). Functional comparison further revealed the defects in cell proliferation and colony formation resulted from the Y119F mutation (Fig. EV5D,E). Together, glycolytic shunts and tumor cell proliferation are compromised upon the disruption of PGAM1 Y119 phosphorylation and the blockage of PGAM1–PKM2 interaction. Next, PGAM1 WT- and Y119F mutant-restored cells were subcutaneously injected into athymic nude mice (Fig. 5E). Both the volume and mass of the tumors developed from PGAM1 Y119F mutant-restored cells significantly decreased compared with that from PGAM1 WT-restored ones (Fig. 5F,G). In line with this observation, PGAM1 Y119 phosphorylation in Y119F mutant tumor xenografts was thoroughly wiped off (Fig. 5H); while Ki-67, a classic marker of cell proliferation, was also decreased due to PGAM1 Y119 mutation (Fig. 5H). These results suggest that PGAM1 Y119 phosphorylation is important for tumor growth.

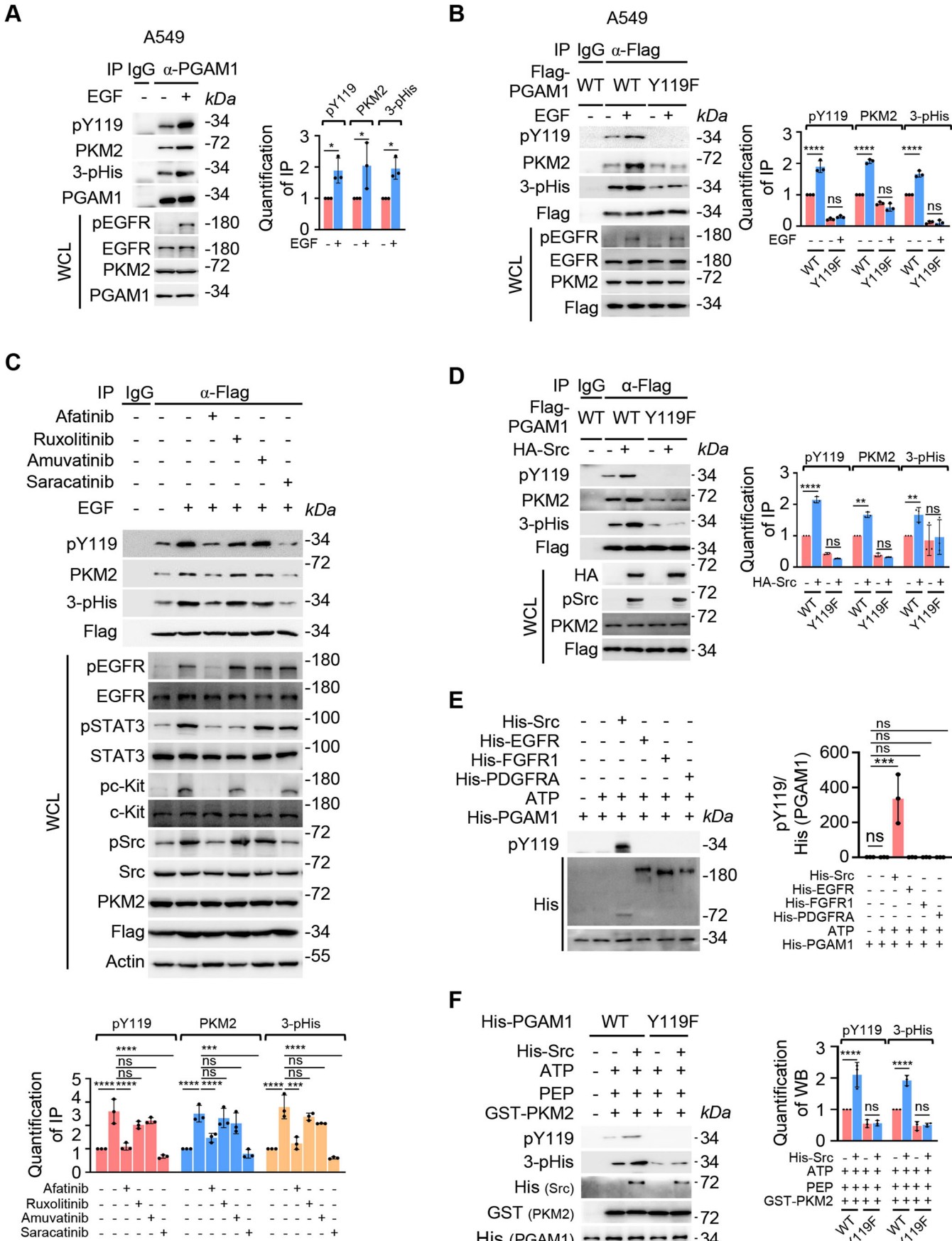

**Figure 4.   EGF signaling accounts for PGAM1 Y119 phosphorylation.**

(A, B) EGF triggers PGAM1 Y119 phosphorylation. Left: A549 cells, or A549 cells transfected with Flag-PGAM1 plasmids were cultured in the serum-free medium for 48 h, and then treated with or without EGF (100 ng/mL) for 30 min. Flag (PGAM1)-associated proteins were immunoprecipitated and analyzed by WB. Right: Quantification of IP. Relative levels of pY119, PKM2 or 3-pHis were normalized to that of PGAM1 or Flag (PGAM1) for each group. (C) Inhibition of EGF-Src signaling decreases PGAM1 Y119 phosphorylation. Top: A549 cells were transfected with Flag-PGAM1 and cultured in serum-free medium for 48 h, and then treated with Afatinib (1 μM), Ruxolitinib (1 μM), Saracatinib (1 μM), or Amuvatinib (10 μM) for 3 h before incubation with EGF (100 ng/mL) for 30 min. Subsequently, Flag (PGAM1)-associated proteins were immunoprecipitated and analyzed by WB. Bottom: Quantification of IP. Relative levels of pY119, PKM2 or 3-pHis were normalized to that of Flag (PGAM1) for each group. (D) Src increases PGAM1 Y119 phosphorylation. Left: A549 cells were co-transfected with Flag-PGAM1 and HA-Src for 48 h. Flag (PGAM1)-associated proteins were immunoprecipitated and analyzed by WB. Right: Quantification of IP. Relative levels of pY119, PKM2 or 3-pHis were normalized to that of Flag (PGAM1) for each group. (E) Src is the kinase for PGAM1 Y119 phosphorylation. Left: In vitro kinase assay was carried out with recombinant His-PGAM1, His-Src, His-EGFR, His-FGFR1, His-PDGFRA in the presence of ATP. PGAM1 Y119 phosphorylation level was detected by WB. Right: Quantification of WB. Relative levels of pY119 were normalized to that of His (PGAM1) for each group. (F) Src phosphorylates PGAM1 at Y119. Left: In vitro kinase assay was carried out with recombinant His-PGAM1, GST-PKM2 and His-Src in the presence of ATP or PEP. PGAM1 H11 phosphorylation level was detected by WB. Right: Quantification of WB. Relative levels of pY119 or 3-pHis were normalized to that of His (PGAM1) for each group. Data information: for WB in (A–F), one representative experiment out of three is shown. For IP in (A–D), IgG served as a negative control. Data were represented as mean ± SD ($n = 3$) with significance determined by one-way (C, E) or two-way ANOVA test (A, B, D, F); ****$P < 0.0001$, ***$P < 0.001$, **$P < 0.01$, *$P < 0.05$, ns nonsignificant. Source data are available online for this figure.

## PGAM1-derived pY119-containing peptide blocks PGAM1–PKM2 interaction and inhibits tumor growth

To verify the role of PGAM1–PKM2 interaction in tumor development, we examined the levels of PGAM1 pY119, PKM2, and Src in tumor tissues versus the adjacent normal tissues from 30 patients (Fig. 6A). The expression of both PKM2 and Src was significantly higher in tumor tissues than that in adjacent normal tissues (Appendix Fig. S4A,B), supporting the functional link between the highly expressed PKM2 and Src and the tumor development (Christofk et al, 2008a; Irby and Yeatman, 2000). According to the data showing PGAM1 Y119 phosphorylation more evident in tumor cells (Figs. 3H and EV3), Y119 phosphorylation in patients' tumor tissues is significantly higher than that in the adjacent normal tissues (Fig. 6B). As expected, so did the level of PGAM1 H11 phosphorylation and PGAM1–PKM2 association (Fig. 6C; Appendix Fig. S4C,D). Notably, the level of PGAM1 Y119 phosphorylation was positively correlated with the expression of both the "writer" Src and the "reader" PKM2 (Fig. 6D,E). More importantly, patients with high levels of PGAM1 pY119 displayed obviously shorter survival duration (average 17.5 months) than those with low level of PGAM1 pY119 (average 37.3 months) (Fig. 6F). Collectively, these results suggest that PGAM1 pY119 phosphorylation regulated by PKM2 specify the tumor tissues from the normal counterparts and signify an unfavorable prognosis.

Based on this specificity of Y119 phosphorylation in tumor cells, a cell-permeable peptide, which was derived from PGAM1 and contained phosphorylated Y119, was designed and fused with HIV-TAT (pY119-TAT) (Fig. 7A). Protein Co-IP assay showed that pY119-TAT peptide, but not TAT alone, interfered with the interaction between PGAM1 and PKM2 in A549 cells (Fig. 7B). To evaluate the therapeutic potential of pY119-TAT peptide in vivo, A549 cells were subcutaneously injected into athymic nude mice (Fig. 7C). Five days after tumor colonization, 10 mg/kg of TAT or pY119-TAT peptide was injected intraperitoneally every other day. Another 10 days after peptide injection, when the tumor xenografts were up to 500 mm³ approximately in the TAT peptide-administrated mice, samples were harvested. Both tumor volume and mass in the mice injected with pY119-TAT peptide were significantly less compared with that injected with TAT peptide (Fig. 7D–F); whereas, the mice's bodyweight had no significant

difference between the two groups (Fig. 7G). Y119 phosphorylation, PKM2 binding, H11 phosphorylation and the enzymatic activity of PGAM1 from the tumor xenografts were further examined, and they were all significantly declined due to pY119-TAT peptide injection (Fig. 7H,I). Collectively, pY119-containing peptide treatment interferes with the interaction of PGAM1–PKM2 and restricts tumor growth.

## Discussion

PGAM1 plays an important role in coordinating glycolysis with its shunts and is crucial for tumor cell proliferation; however, the comprehension of the functional regulation of this key metabolic enzyme still remains elusive. In the present study, we reveal that PKM2, conventionally transferring the phosphate from PEP to ADP and producing pyruvate and ATP, functions as a histidine kinase, transferring phosphate from PEP to PGAM1. PKM2 phosphorylates PGAM1 at H11 and this unique posttranslational modification elevates the enzymatic activity of the latter, thereby enhances glycolysis shunts and promotes tumor growth.

As the rate-limiting enzyme controlling the final step of glycolysis, pyruvate kinases (PKs) are comprised of four isoenzymes (M1, M2, L, and R), that all catalyze ADP and PEP into ATP and pyruvate (Chaneton and Gottlieb, 2012b; Dayton et al, 2016; Luo and Semenza, 2012). While PKL, PKR, and PKM1 are constitutively active as tetramers, PKM2 can selectively form the less active dimers or inactive monomers other than the metabolically active tetramers, to regulate variable metabolic demands (Dombrauckas et al, 2005; Morgan et al, 2013). Notably, PKM2 is the dominant PK in highly proliferative cells including cancer cells, and is pivotal for the Warburg effect that is required for tumor cell proliferation (Christofk et al, 2008a; Sun et al, 2011; Ye et al, 2012). We and others previously demonstrated that PKM2 de-tetramerization was induced by its *O*-GlcNAcylation, phosphorylation or acetylation which impinged on nuclear (nonmetabolic) functions of PKM2 to promote the Warburg effect (Lv et al, 2013; Pan et al, 2021; Wang et al, 2017; Yang and Lu, 2013). PKM2 monomer and dimer translocated into nucleus, phosphorylated histone H3 and stimulated c-Myc-dependent expression of key glycolytic components, such as GLUT1 and LDHA (Wang et al, 2017; Yang et al, 2011; Yang et al, 2012b). In the present study, it is also monomeric and

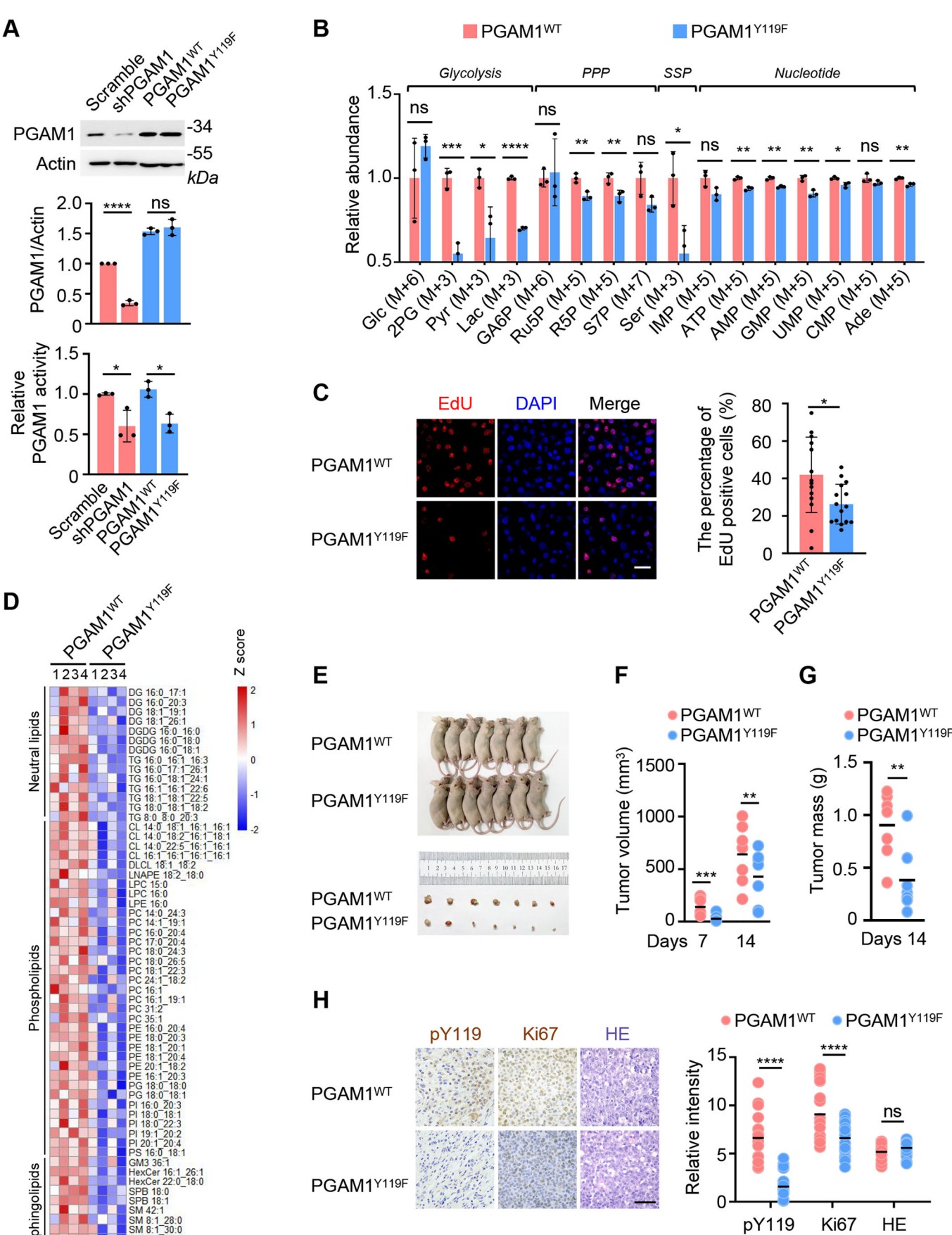

**Figure 5.   Glycolytic shunts and cell proliferation are compromised due to the mutation of PGAM1 Y119.**

(A) Establishment of PGAM1 rescued cell lines. Endogenous PGAM1 was depleted from A549 cells, and then PGAM1 WT or Y119F mutant were restored and stably expressed. Top: PGAM1 expression was analyzed by WB. Middle: Relative levels of PGAM1 were normalized to that of Actin for each group. Bottom: PGAM1 activity were examined by commercially available kits. $n = 3$ biological replicas. (B) Impact of PGAM1 Y119 mutation on glucose metabolism flux. PGAM1 WT- or Y119F mutant-restored cells were cultured for 24 h in the presence of [$^{13}C_6$] glucose (25 mM) and then subjected to metabolic flux assay by ultra-high performance liquid chromatography-high resolution mass spectrometry (UHPLC-HRMS). $n = 3$ biological replicas. The major metabolites in glycolysis, PPP, SSP and nucleotide metabolism is shown. Glc glucose, 2PG 2-phosphoglycerate, Pyr pyruvate, Lac lactate, GA6P gluconic acid-6-phosphate, Ru5P ribulose-5-phosphate, R5P ribose-5-phosphate, S7P sedumketose-7-phosphate, Ser serine, IMP inosine monophosphate, ATP adenosine triphosphate, AMP adenosine monophosphate, GMP guanosine monophosphate, UMP uridine monophosphate, CMP cytidine monophosphate, Ade adenosine. (C) PGAM1 Y119 mutation decreases DNA replication. PGAM1 WT- and Y119F mutant-restored cells were pulse-labeled with EdU and analyzed for DNA synthesis by immunofluorescence. Representative images (left) and associated quantifications (right) were displayed. Scale bars, 25 μm. $n = 15$, five random fields from each of the three experiments. (D) PGAM1 Y119 mutation reduces lipid levels. Lipid levels in PGAM1 WT- and Y119F mutant-restored cells were analyzed by UHPLC-HRMS and filtrated by FC > 1.2 and $P < 0.05$. z-score was calculated from the average value of each group ($n = 4$). z-score> 0 and <2 indicate a significant increase (red), and z-score <0 and >-2 indicate a significant decrease (blue). (E–G) PGAM1 Y119 mutation compromises tumor formation in nude mice. PGAM1 WT- and Y119F mutant-restored cells were respectively injected into athymic nude mice. The xenograft tumors were sampled and photographed after 14 days. (E) Mice-bearing tumor and the tumors dissected from the mice were photographed. (F) Tumor volumes were measured according to length (l) and width (w) and calculated by the equation $v = lw^2/2$ ($n = 7$). (G) The average mass of xenograft tumors was quantified ($n = 7$). (H) Immunohistochemistry (IHC) analysis of tumor cell proliferation. Seven pairs of mice respectively injected with PGAM1 WT- and Y119F mutant-restored cells were co-stained with Ki-67 and hematoxylin and eosin (HE). Left: Representative images depicting tumor tissues. Scale bars, 50 μm. Right: Relative intensity of pY119, Ki-67 and HE staining ($n = 21$, three random fields from each of the seven analyzed mice). Data information: data are represented as mean ± SD ($n \geq 3$) with significance determined by one-way ANOVA test (A) or Student's $t$ test (B–G); ****$P < 0.0001$, ***$P < 0.001$, **$P < 0.01$, *$P < 0.05$, ns nonsignificant. Source data are available online for this figure.

dimeric PKM2 that were more efficient to phosphorylate and activate PGAM1 to augment glycolytic shunts, while the tetrameric PKM2 was not. This is quite reasonable because the tetramerization of PKM2 buries the appropriate interface that is accommodated for the substrate proteins' docking. This not only broadens the understanding about the nonmetabolic function of PKM2 dimers and monomers, but also suggests another advantage of PKM2 de-tetramerization for Warburg effect which promoted the anabolic pathways by upregulating PGAM1 activity beyond causing the passive accumulation of glycolytic intermediates towards to glycolytic shunts due to the reduced PK activity as previously proposed (Anastasiou et al, 2011; Lv et al, 2011; Wang et al, 2017). Importantly, the present study also proposes a metabolic-enzyme-crosstalk-based regulatory mode for metabolic reprogramming.

It has been documented that dimeric PKM2 is involved in threonine phosphorylation of histone H3, tyrosine phosphorylation of transcriptional factor STAT3 and serine phosphorylation of synaptosome-associated protein 23 (Gao et al, 2012; Wei et al, 2017; Yang et al, 2012a). While these previous studies declared that PKM2 use PEP as phosphodonor to function as a serine/threonine/tyrosine protein kinase, the present study shows that PKM2 monomer and dimer efficiently phosphorylated H11 of PGAM1, identifying PKM2 as a histidine kinase. Histidine phosphorylation, originally discovered in prokaryotes and has recently been successively identified in eukaryotes, is important for many cellular events including signal transduction, cell cycle progression and phagocytosis (Fuhs and Hunter, 2017; Hunter, 2022; Potel et al, 2018; Potel et al, 2019). However, to date, only two histidine kinases have been identified in mammalian cells, namely NME1 and NME2 from the nucleoside diphosphate kinases family (Adam and Hunter, 2018; Attwood and Wieland, 2015). NME1/2, the metabolic enzymes that conventionally transfer the phosphate from NTP to NDP, moonlight as histidine kinases by using their metabolic substrate ATP or GTP as a phosphodonor (Adam and Hunter, 2018; Attwood and Wieland, 2015). NME1 phosphorylates the histidine at the catalytic site of ATP citrate lyase (Wagner and Vu, 1995), a cytosolic metabolic enzyme that is critical for fatty acid synthesis; NME2 can form a complex with G protein βγ dimers and

phosphorylate H266 in the β subunit of G protein complex (Cuello et al, 2003). This study revealed that PKM2 uses its metabolic substrate PEP to catalyze PGAM1 H11 phosphorylation. Studies of ours and others suggested a regulation paradigm that metabolic kinases may switch to histidine kinases by utilizing their cognate metabolic substrates as phosphodonors. Nevertheless, whether this PKM2-mediated PEP-consuming histidine phosphorylation may apply to a wide range of receptor proteins other than PGAM1 is an urgent question to be answered.

In this work, through unveiling the glycolytic enzyme PKM2-mediated PGAM1 H11 phosphorylation, we identify a unique histidine kinase in tumor cells that exploits PEP as phosphodonor. A previous study documented that decreased pyruvate kinase activity allows PEP-dependent H11 phosphorylation of PGAM1, providing insight into how an alternative glycolytic pathway decouples ATP production from PEP-based phosphotransferring and leads to a high rate of glycolysis to support the anabolic metabolism in many proliferating cells (Vander Heiden et al, 2010). To identify the kinase(s) controlling of PGAM1 H11 phosphorylation, the authors took into account the potential involvement of PKM1 and 2; however, co-depletion of both PKM1 and PKM2 did not significantly impair PGAM1 H11 phosphorylation in HEK-293T cells (Vander Heiden et al, 2010). In this study, in both lung cancer cell A549 and breast cancer cell MCF-7, we observed a significant reduction of PGAM1 H11 phosphorylation upon PKM2-specific depletion (Figs. 1G and EV1E). We speculate that, in contrast to the removal of PKM2 alone, PKM1/2 co-depletion might result in an over accumulation of PEP, causing a spontaneous phosphorylation at PGAM1 H11 (Vander Heiden et al, 2010). Indeed, PGAM1 was spontaneously phosphorylated independent of PKM2 when we artificially elevated PEP levels higher than millimolar magnitudes in the in vitro kinase assay (Appendix Fig. S5). This may be due to the favor of the high energy phosphate bond in PEP, and may provide an explanation for PKM1/2 co-depletion-caused enhancement of PGAM1 H11 phosphorylation. Our work not only supports the conclusion that PEP-dependent PGAM1 H11 phosphorylation occurs in PKM2-expressing tumor cells (Vander Heiden et al, 2010) but also

                                                                          

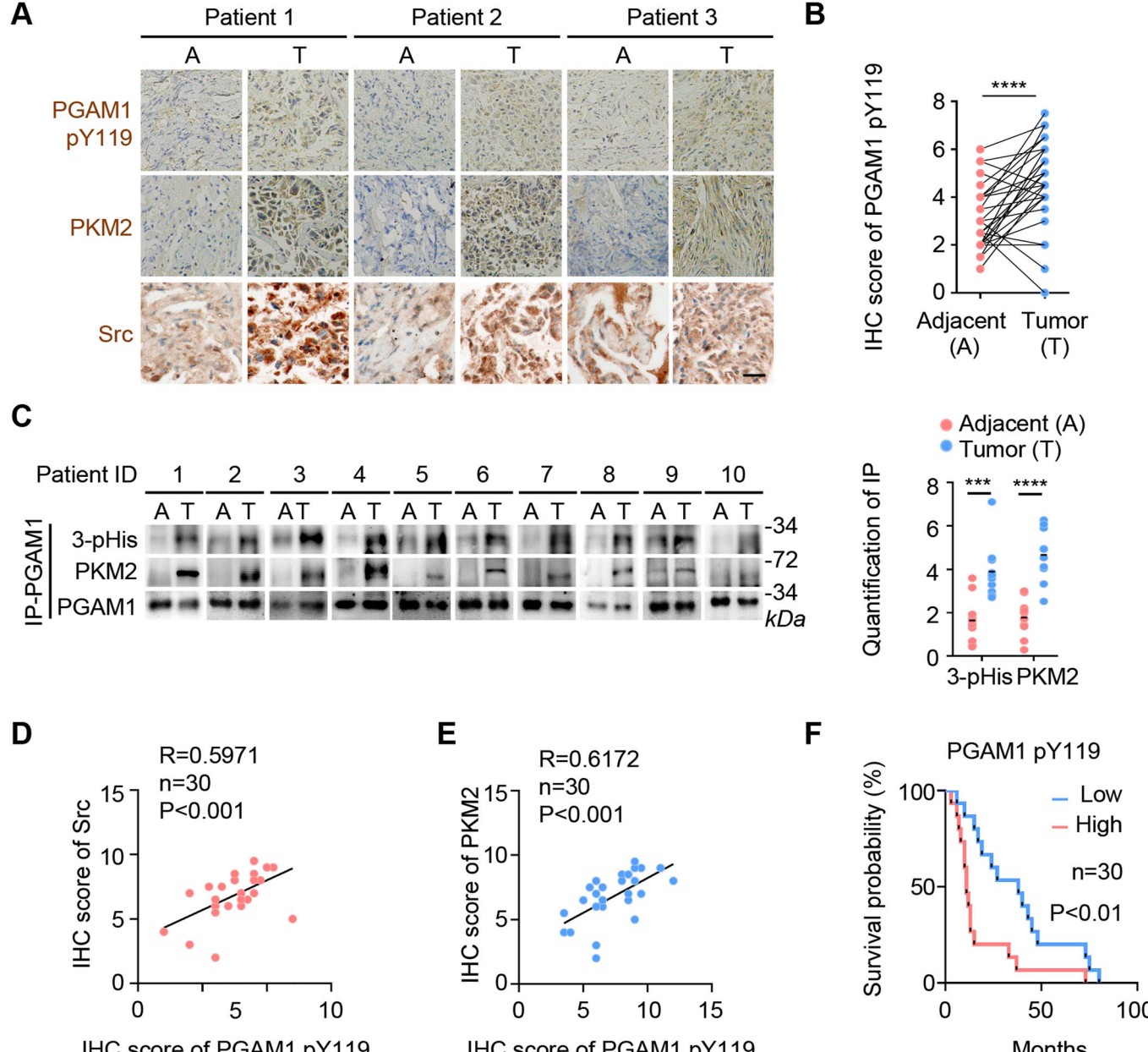

**Figure 6. PGAM1 Y119 phosphorylation specify the tumor tissues and signify an unfavorable prognosis.**

(A) IHC analysis of the levels of PGAM1 pY119, Src and PKM2 in human cancer tissues and adjacent normal tissues. Tumor tissues and adjacent normal tissues from 30 patients were stained with antibodies against PGAM1 pY119, Src and PKM2. Representative images of three tumors are shown. Scale bars, 25 μm. (B) PGAM1 pY119 level in tumor tissues is higher than that in adjacent normal tissues. The levels of pY119 from (A) were scored and subjected to statistical analysis. $n = 30$ pairs of adjacent and tumor tissues. Data are represented as mean ± SD with significance determined by Student's $t$ test, ****$P < 0.0001$. (C) PGAM1 pH11 level in tumor tissues is higher than that in adjacent normal tissues. Left: PGAM1-associated proteins from tumor tissues and adjacent normal tissues of ten patients were immunoprecipitated and analyzed by WB. Right: Quantification of IP. Relative levels of 3-pHis or PKM2 were normalized to that of PGAM1 for each group. $n = 10$ pairs of adjacent and tumor tissues. Data are represented as mean ± SD with significance determined by Student's $t$ test, ****$P < 0.0001$, ***$P < 0.001$. (D, E) PGAM1 pY119 level is positively correlated with levels of PKM2 or Src. PGAM1 pY119, PKM2 and Src staining from (A) were scored and subjected to correlation analysis. The correlation analysis was determined by Pearson product moment correlation test ($n = 30$), $P < 0.001$. (F) Survival duration analysis. Survival duration of 30 lung cancer patients with low (15 cases, blue curve) versus high (15 cases, red curve) PGAM1 pY119 from (A) was compared and determined by two-tailed log-rank test ($n = 30$), $P < 0.01$. Source data are available online for this figure.

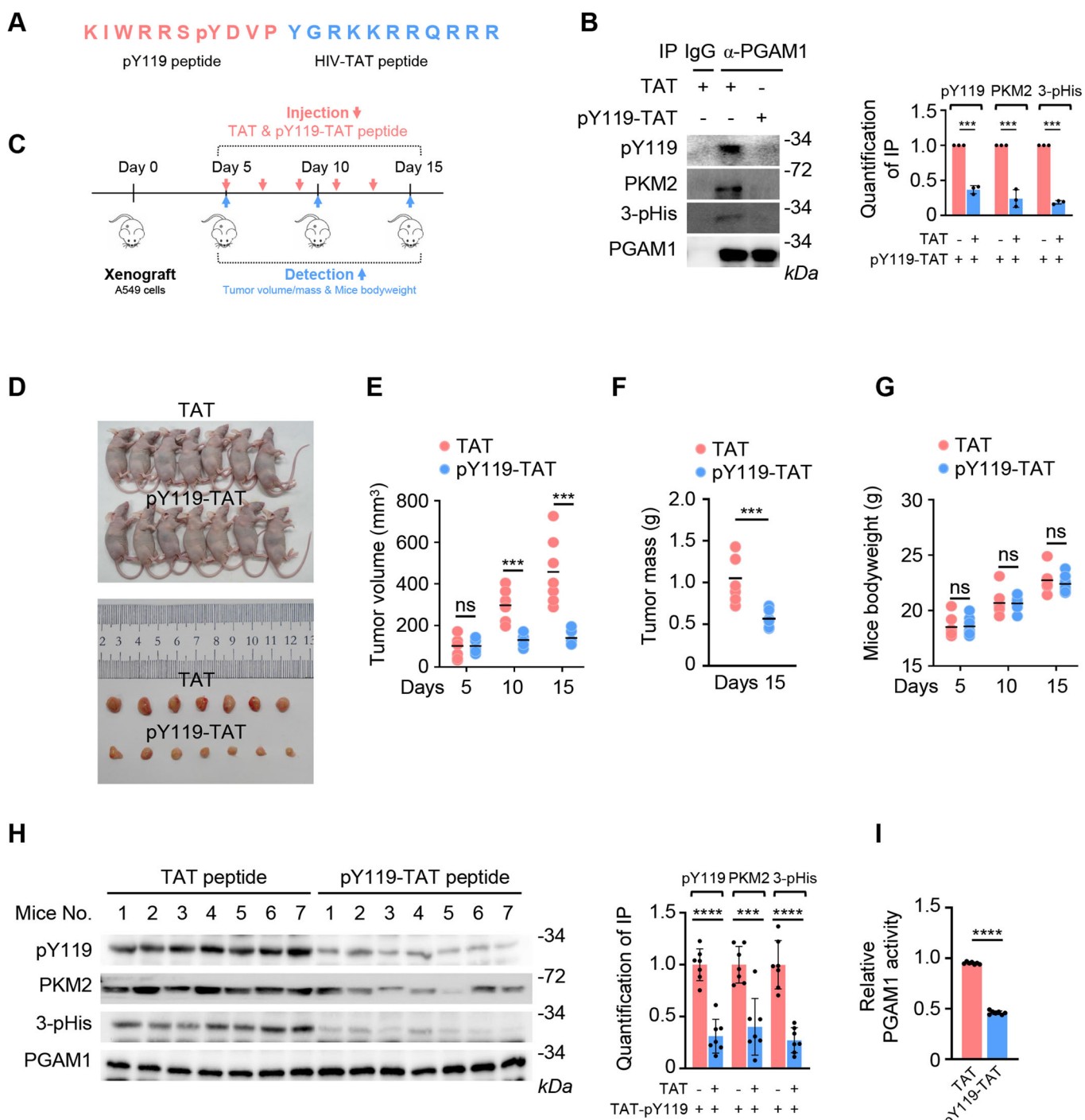

**Figure 7. PGAM1-derived pY119-containing peptide blocks PGAM1–PKM2 interaction and inhibits tumor growth.**

(**A**) Design of pY119-TAT peptide. PGAM1 sequence contained pY119 (amino acid residues 113–122, red letters) were fused with HIV-TAT peptide (blue letters). (**B**) pY119-TAT peptide reduces PGAM1–PKM2 association. Left: A549 cells were treated with TAT peptide (25 μM) or pY119-TAT peptide (25 μM) for 24 h. PGAM1-associated proteins were immunoprecipitated and analyzed by WB. One representative experiment out of three is shown. IgG served as a negative control. Right: Quantification of IP. Relative levels of pY119, PKM2 or 3-pHis were normalized to that of PGAM1 for each group. (**C–G**) pY119-TAT peptide harnesses tumor formation in nude mice. (**C**) A schematic diagram of the xenograft experiment. A549 cells were injected into athymic nude mice. Five days after tumor colonization, 10 mg/kg of TAT peptide or pY119-TAT peptide was injected intraperitoneally every other day. The xenograft tumors were sampled and photographed after 15 days. (**D**) Mice-bearing tumor and the tumors dissected from the mice were photographed. Tumor volume (**E**), tumor mass (**F**), and mice bodyweight (**G**) were measured (*n* = 7). (**H**, **I**) pY119-TAT peptide significantly decreased PGAM1 enzymatic activity. The PGAM1 proteins were immunoprecipitated from the xenograft tumors tissues in (**D**), and then subjected to WB analysis (**H**) or enzymatic activity analysis (**I**). *n* = 7 pairs of xenograft tumors tissues. Data information: data were represented as mean ± SD (*n* ≥ 3) with significance determined by Student's *t* test (**B**, **E–I**); ****$P$ < 0.0001, ***$P$ < 0.001, ns nonsignificant. Source data are available online for this figure.

provides a more detailed molecular mechanism for PEP-based phosphotransferring to PGAM1 when pyruvate kinase activity of PKM2 is reduced.

It has been documented that, as a phosphotyrosine-binding protein, PKM2 binds with β-catenin at its phosphorylated Y333 in the nucleus (Yang et al, 2011). Our previous study also depicted that PKM2 recognizes and associates with OGT that is phosphorylated at Y976, which in turn lead to O-GlcNAcylation of PKM2 (Wang et al, 2022). In the present study PGAM1 Y119 phosphorylation is identified as a prerequisite for the recognition of PKM2, suggesting that the phosphorylated tyrosine of the substrate proteins may be a universal requirement for recognition by PKM2. The level of PGAM1 pY119 phosphorylation in tumor cells or tissues is significantly higher than that of untransformed cells or tumor-adjacent tissues; accordingly, PGAM1 H11 phosphorylation was co-related Y119 phosphorylation. PGAM1 Y119 phosphorylation is triggered by growth factors' signaling, that is usually constitutively activated in many cancers and important for tumor growth. Thus, the identification of EGF signaling-triggered PGAM1 Y119 phosphorylation not only interprets the molecular mechanism for PKM2 recognition, but also ideally specifies the occurrence of the phosphorylation of both Y119 and H11 in tumor cells, providing a promising target for cancer therapy that may leave the normal proliferating cells unaffected. A PGAM1-derived pY119-containing peptide has been synthesized, which indeed interfers with the interaction of PGAM1 and PKM2, and significantly inhibits tumor growth in mice.

In summary, this study not only demonstrates PKM2 function as a histidine kinase to phosphorylate PGAM1 to coordinate the glycolysis with its shunts during the rapid proliferation of the tumor cells, but also illustrates an enzyme-crosstalk-based regulatory mode during metabolic reprogramming. From a clinical perspective, targeting PKM2–PGAM1 regulatory metabolic joint may hold promise for the development of strategies for cancer therapies.

# Methods

## Cell culture

Human A549 lung cancer cells were cultured in Ham's F-12K (Sigma, Saint Louis, MO, USA) supplemented with 10% FBS (Procell, Wuhan, China) and 1% penicillin/streptomycin (Sigma). Human HBE-2 bronchial epithelial cells, human H1299 lung cancer cells, human MCF-7 breast cancer cells, human gastric mucosal epithelial cell line GES-1, human AGS gastric cancer cells, and human embryonic kidney 293T (HEK-293T) cells were cultured in DMEM medium (Sigma) supplemented with 10% FBS and 1% penicillin/streptomycin. T lymphocyte leukemia Jurkat cells were cultured in RPMI-1640 medium (Sigma) supplemented with 10% FBS and 1% penicillin/streptomycin. Human MCF-10A immortalized mammary epithelial cells were cultured in MCF-10A medium (Procell). Human MDA-MB-231 breast cancer cells were cultured in L15 medium (Sigma) supplemented with 10% FBS and 1% penicillin/streptomycin. All cell lines were purchased from the Cell Bank of Type Culture Collection of the Chinese Academy of Sciences and authenticated by the distributors. All cells were negative of mycoplasma contamination.

## Human T-cell isolation and activation

Naive T cells were purified from whole blood of healthy human subjects based on immunomagnetic separation by AutoMACS System (Miltenyi, Bergisch Gladbach, Germany). The blood was obtained from Jilin Blood Center (Changchun, China). Isolated cells were plated at a density of $1 \times 10^6$ cells/mL in RPMI-1640 medium, supplemented with 10% FBS and 1% penicillin/streptomycin. Activation and expansion of T cells were performed using 30 U/mL of recombinant human IL-2 (Invitrogen, Carlsbad, CA, USA) and Dynabeads Human T Activator CD3/CD28 (Invitrogen) according to the manufacturer's recommendation. Following the 72 h-incubation in a humidified $CO_2$ incubator at 37 °C, activated cells were confirmed by flow cytometry (FACSCanto™II, BD, Franklin Lake, NJ, USA). The present study was approved by the Ethical Committee of Northeast Normal University (202302036, Changchun, China) and conducted with the informed consent of all donors.

## Reagents

DNA transfection reagent lipofectamine 2000 was purchased from Invitrogen. Recombinant human EGF was purchased from R&D Systems (Minneapolis, MN, USA). Recombinant proteins, including human FGF1, PDGFA, EGFR, FGFR, and PDGFRA, were purchased Abclonal. 2,3-BPG, PEP, ATP, GTP were purchased from Sigma. Inhibitors, including Afatinib, ruxolitinib, amuvatinib, saracatinib, zoligratinib (CH5183284) and crenolanib were purchased from Selleck (Shanghai, China). TEPP46 was purchased from MedChemExpress (Monmouth Junction, NJ, USA). Recombinant His-tagged NME1 protein was purchased from Solarbio (Beijing, China). Recombinant NME2 protein was purchased from Abcam (Cambridge, UK). Pronase was purchased from Sigma.

## DNA constructs and mutagenesis

PCR-amplified human PGAM1, PKM2, PKM1, ENO1 and PTP1B cDNA were cloned into p3 × Flag-CMV-10, pcDNA3.1, pET-28a, pGEX-6P-1 or pCDH-CMV vectors. Mutations, including PGAM1$^{H11N}$, PGAM1$^{Y26F}$, PGAM1$^{Y50F}$, PGAM1$^{Y92F}$, PGAM1$^{Y119F}$, PGAM1$^{Y133F}$, PGAM1$^{Y142F}$, PGAM1$^{Y26E}$, PGAM1$^{Y50E}$, PGAM1$^{Y92E}$, PGAM1$^{Y119E}$, PGAM1$^{Y133E}$, PGAM1$^{Y142E}$, PKM2$^{Y105E}$, PKM2$^{R399E}$, PKM2$^{K422R}$, PKM2$^{K270M}$, PKM2$^{Y105E/K270M}$, PKM2$^{R399E/K270M}$ and PKM2$^{K422R/K270M}$ were made using the QuickChange Site-directed Mutagenesis Kit (Takara, Beijing, China). Original pcDNA3.1-HA-Src plasmid was generously provided by Dr. Weiwei Yang (Shanghai Institute of Biochemistry and Cell Biology, Chinese Academy of Sciences). We subsequently constructed Src into pET-28a vector.

## Stable cell line generation

For PKM2- or PGAM1-depleted cell lines, pLVTHM PKM2 shRNA was generated with CATCTACCACTTGCAATTA oligonucleotide targeting exon 10 of the PKM2 transcript. pLVTHM PGAM1 shRNA was generated with CGCCTCAATG AGCGGCACTAT oligonucleotide targeting PGAM1 transcript. The pLVTHM control was generated with control oligonucleotide GCTTCTAACACCGGAGGTCTT. PKM2 or PGAM1-depleted cell

line was generated by lentiviral transduction in A549 and MCF-7 cells. Briefly, lentivirus particles were generated in HEK-293T cells by co-transfection of the plasmid psPAX and Pmd2.G along with either pLVTHM-shPKM2 or -shPGAM1. Virus supernatant supplemented with polybrene (Sigma) was used to infect A549 or MCF-7 cells. Infected cells were then subjected to medium containing puromycin (2 μg/mL, Sigma) for 1 week' selection prior to the expression test.

For PKM2 or PGAM1 rescue cell lines, wild type or mutant pCDH-PKM2 or -PGAM1 plasmids along with psPAX and Pmd2.G were co-transfected into HEK-293T to prepare the lentiviruses. Cells depleted for PKM2 or PGAM1 were infected with lentiviruses containing PKM2 or PGAM1 in the presence of polybrene and selected with puromycin (2 μg/mL, Sigma) and hygromycin (400 μg/mL, Sigma) for 2 weeks.

## Quantitative real-time PCR

Total RNA was isolated using the TRIzol reagent (Invitrogen). Complementary DNA (cDNA) was synthesized with the PrimeScript RT reagent Kit plus gDNA Eraser (Takara) according to the manufacturer's instructions. Quantitative real-time PCR (pPCR) was performed on the QuantStudio 3 Real-Time PCR Instrument (Applied Biosystems) with a TB Green Premix Ex Taq (Tli RNaseH Plus) reagent (Takara). The mRNA expression of genes was normalized to the expression of β-actin gene. Data were analyzed using the comparative cycling threshold method. pPCR primers used in this study include *ENO1*, forward 5'-AAAGCTGGTGCCGTTGAGAA-3', reverse 5'-GGTTGTGGTAAACCTCTGCTC-3'; *ENO2*, forward 5'-AGCCTCTACGGGCATCTATGA-3', reverse 5'-TTCTCAGTCCCATCCAACTCC-3'; *ENO3*, forward 5'-GGCTGGTTACCCAGACAAGG-3', reverse 5'-TCGTACTTCCCATTGCGATAGAA-3'; *ENO4*, forward 5'-AGCTGTGGGACCACTAGGG-3', reverse 5'-TATGTCCACTTGCAGGGTTGG-3'; *PKM*, forward 5'-ATGTCGAAGCCCCATAGTGAA-3', reverse 5'-TGGGTGGTGAATCAATGTCCA-3'; *PKLR*, forward 5'-TCAAGGCCGGGATGAACATTG-3', reverse 5'-CTGAGTGGGGAACCTGCAAAG-3'; *PCK1*, forward 5'-TTGAGAAAGCGTTCAATGCCA-3', reverse 5'-CACGTAGGGTGAATCCGTCAG-3'; *PCK2*, forward 5'-AGTAGAGAGCAAGACGGTGAT-3', reverse 5'-TGCTGAATGGAAGCACATACAT-3'.

## Immunoprecipitation and western blot

Cells were lysed in cold WB-IP lysis buffer (Beyotime, Shanghai, China) for 30 min and centrifuged ($12,000 \times g$ for 15 min at 4 °C) to remove cell debris. Supernatants were collected and used as whole-cell lysates. The proteins were immunoprecipitated from the whole-cell lysates using indicated antibodies coupling to Protein A/G PLUS Agarose beads (Santa Cruz, CA, USA) for 3 h at 4 °C. Nonspecific mouse IgG antibody coupling to Protein A/G PLUS Agarose beads was used as a negative control. For PGAM1 H11 phosphorylation assay, immunoprecipitated PGAM1 proteins on the beads were competitively eluted by poly peptides derived from Flag or PGAM1, and then prepared in SDS-PAGE loading buffer (5× = 10% SDS, 250 mM Tris-HCl, pH 8.8, 0.02% bromophenol blue, 50% glycerol, 50 mM EDTA, 500 mM DTT) and analyzed by western blot modified for "PGAM1 H11 phosphorylation" section (see below). Except for PGAM1 H11 phosphorylation assay,

immunoprecipitated proteins on the beads were directly boiled off with SDS-PAGE loading buffer (pH 6.8) for 5 min and analyzed by western blot with standard procedures.

The primary antibodies used in this study include anti-PGAM1 (1:200 for WB and 1:10 for IP, Santa Cruz, 130334), anti-PKM2 (1:1000 for WB and 1:50 for IP, CST, 4053 S), anti-PKM1 (1:1000 for WB, CST, 7067), anti-NME1 (1:1000 for WB, Abclonal, A8802), anti-NME2 (1:1000 for WB, Abclonal, A7443), anti-phospho-tyrosine (pTyr) (1:1000 for WB, CST, 8954), anti-Flag (1:1000 for WB and 1:50 for IP, Sigma, F1804), anti-Actin (1:1000 for WB, Sigma, A4700), anti-ENO1 (1:1000 for WB and 1:20 for IP, CST, 3810), anti-PCK2 (1:1000 for WB, SAB, College Park, MD, USA, A11526), anti-His (1:1000 for WB, TransGen, 32301) and anti-GST (1:1000 for WB, TransGen, ab2739), anti-HA (1:1000 for WB, TransGen, HT301-01), anti-3-pHis (1:1000 for WB, Millipore, MABS1351), anti-Tubulin (1:1000 for WB, TransGen, HC101-01), anti-EGFR (1:1000 for WB, Abclonal, A23452), anti-pEGFR (1:1000 for WB, SAB, 11220), anti-STAT3 (1:1000 for WB, CST, 9139S), anti-pSTAT3 (1:1000 for WB, CST, 9145S), anti-c-Kit (1:1000 for WB, Abclonal, A0357), anti-pc-Kit (1:1000 for WB, Abclonal, AP0586), anti-Src (1:1000 for WB, Abclonal, A11707), anti-pSrc (1:1000 for WB, Abclonal, AP1377), anti-FGFR1 (1:1000 for WB, Abclonal, A21219), anti-pFGFR1 (1:1000 for WB, Abclonal, AP1317), anti-PDGFRA (1:1000 for WB, Abclonal, A2103), anti-pPDGFRA (1:1000 for WB, Abclonal, AP0615), anti-Ki-67 (1:100 for IHC, CST, 34330). PGAM1 pY119 site-specific antibody (1:1000 for WB and 1:100 for IHC) was generated from Abclonal Technology (Wuhan, China). The PGAM1 phosphopeptide KIWRRS(p)YDVP and non-phosphopeptide KIWRRSYDVP were synthesized and conjugated with KLH, respectively. The serum antibody was produced by immunizing rabbits with synthetic phosphopeptide KIWRRS(p)YDVP after five immunizations. Non-phospho-specific antibodies were removed by chromatography using non-phosphopeptide KIWRRSYDVP. PGAM1 pY119 site-specific antibody was purified by affinity-chromatography using epitope-specific KIWRRS(p)YDVP phosphopeptide. In the following validation assays, we have thoroughly verified the specificity of the PGAM1 pY119 polyclonal antibody. The secondary antibodies included goat anti-mouse IgG secondary antibody HRP-conjugated (SAB, L3032, 1:5000) and goat anti-rabbit IgG secondary antibody HRP-conjugated (SAB, L3012, 1:5000). Mouse IgG (Sigma, I8765, 1:50) and Rabbit IgG (Sigma, I8140, 1:50).

## Protein expression and purification

PKM2, PKM1, and ENO1 were subcloned into the pGEX-6P-1 GST-fusion vector. BL21 (DE3) cells containing pGEX-PKM2, pGEX-PKM1 or pGEX-ENO1 were cultured from single colonies and grown at 37 °C for 16 h in LB medium broth supplemented with 100 μg/mL ampicillin with shaking at 180 rpm. Expression cultures were diluted from starter cultures with the same medium to OD = 600 of 0.2. Protein expression was induced with 1 mM IPTG (Sigma) at OD = 600 of 0.6 for 3 h at 30 °C. Bacteria were pelleted ($12,000 \times g$ for 10 min at 4 °C) and resuspended in 1 mL GST lysis/wash buffer (150 mM NaCl, 20 mM HEPES, 5 mM MgCl$_2$, 1% Triton-X100, 1 mM PMSF, pH 8.0)/50 mL culture. Lysates were sonicated on ice and clarified by centrifugation

(12,000 × g for 30 min at 4 °C). Glutathione resin was equilibrated with GST lysis/wash buffer and 1 mL washed resin/200 mL culture was incubated with clarified bacterial lysates for 2 h at 4 °C. Resin was then pelleted and the supernatant was removed before washing at least 3 times with 10 mL wash buffer. The washed resin was competitively eluted by glutathione buffer (2.5 mM Tris-HCl, 10 mM KCl, 40 mM glutathione, pH 8.0).

PGAM1, Src and PTP1B were subcloned into the pET-28a 6× His-fusion vector. The expression of His-tagged PGAM1, Src, and PTP1B proteins were induced in bacteria with the same methods of GST-tagged PKM2 and ENO1. Bacteria were pelleted and resuspended in 1 mL His lysis/wash buffer (50 mM NaH$_2$PO$_4$, 300 mM NaCl, 10 mM imidazole, 1 mM PMSF, pH 8.0)/50 mL culture. Lysates were sonicated on ice and clarified by centrifugation. Ni resin was equilibrated with His lysis/wash buffer, and 1 mL washed resin/200 mL culture was incubated with clarified bacterial lysates for 2 h at 4 °C. Resin was then pelleted and the supernatant was removed before washing at least three times with 10 mL wash buffer. The washed resin was competitively eluted by imidazole buffer (50 mM NaH$_2$PO$_4$, 300 mM NaCl, 250 mM imidazole, pH 8.0). All the purified proteins were dialyzed in PBS overnight at 4 °C, and then quantified using a BSA standard curve and analyzed by western blot.

### Biolayer interferometry (BLI) assay

His-tagged PGAM1 (50 μg) purified from *E. coli* were immobilized onto a Ni-NTA biosensor (Pall ForteBio) that had been equilibrated in running buffer [20 mM HEPES (pH 7.5), 150 mM NaCl, and 0.02% Tween-20] for 3 min to remove unspecific bound-proteins. Subsequently, the biosensor coated with His-PGAM1 was transferred into GST-PKM2 (0, 10, 20, 40, 80, 160 μg) containing buffer or analyte-free buffer for the detection of PGAM1–PKM2 association. To examine the disassociation of PGAM1–PKM2, biosensors were inserted into wells filled with only assay buffer. Data were analyzed using Octet System Data Analysis software 8.11 (Pall ForteBio).

### Isothermal titration calorimetry (ITC) measurements

ITC assays were performed as described previously (Zhao et al, 2022) and carried out on a MicroCal PEAQ-ITC calorimeter (Malvern Paralytical) at 25 °C. PBS was used to dilute proteins and 2,3-BPG. The concentration of recombinant His-tagged PGAM1 proteins was 20 μM, and 2,3-BPG was diluted in PBS to 500 μM. The ITC experiments involved 19 injections of chemical compound into protein to detect any heat change. A reference measurement (PBS) was carried out to compensate for the dilution of 2,3-BPG. Curve fitting to a single binding site model was performed using the ITC data analysis module of MicroCal Software provided by the manufacturer.

### Drug-affinity responsive target stability (DARTS) assays

DARTS was performed as described previously (Zhao et al, 2022). Briefly, A549 cells overexpressed with Flag-PGAM1 or different mutants were harvested in cold lysis buffer (Beyotime) with protease inhibitor cocktail (Beyotime), then centrifuged at 13,000 rpm for 15 min. The supernatants were collected and

incubated with varying concentrations of 2,3-BPG or PBS, kept at room temperature for 30 min. Samples were digested with pronase (12.5 μg/mL) for 10 min at room temperature, followed by the addition of SDS-PAGE loading buffer to stop the reaction. Tubulin was used as a negative control.

### In vitro kinase assay

Recombinant His-PGAM1 (10 μg/mL) purified bacterially was incubated with recombinant GST-PKM2/His-PKM2 (20 μg/mL), GST-PKM1 (20 μg/mL), GST-ENO1 (20 μg/mL), His-NME1 (4 μg/mL), NME2 (4 μg/mL), His-Src (6 μg/mL), His-EGFR (20 μg/mL), His-FGFR1 (20 μg/mL) or His-PDGFRA (20 μg/mL) in phosphorylation reaction buffer [50 mM Tris-HCl (pH 8.8), 100 mM KCl, 50 mM MgCl$_2$, 1 mM Na$_3$VO$_4$, 1 mM PMSF, and 1 mM DTT]. 0.5 mM ATP, 0.5 mM PEP or 0.5 mM GTP was added in the kinase buffer to start the reaction. The reactions were performed in a total volume of 100 μL at 30 °C for 30 min and terminated by adding SDS-PAGE loading buffer (pH 8.8) and avoided boiling. The reaction mixtures were then subjected to western blot analysis.

### PGAM1 H11 phosphorylation assay

As previously reported (Fuhs et al, 2015), 3-pHis antibody (Millipore, MABS1351) that recognizes imidazole nitrogen atom (N3) phosphorylation of histidine was used to detect PGAM1 H11 phosphorylation. Briefly, buffers were adjusted to pH 8-9 to stabilize pHis and methods were modified to avoid heating samples. For in vitro kinase assays, recombinant PGAM1 protein samples were prepared in SDS-PAGE loading buffer (pH 8.8) and analyzed immediately using western blot. For the immunoprecipitation assay, cells were washed with cold TBS buffer (pH 8.8) and scraped directly into IP-WB lysis buffer (pH 8.8). The whole-cell lysates were incubated on ice and clarified by centrifugation (12,000 × g for 15 min at 4 °C). Immunoprecipitated PGAM1 proteins from the whole lysates were competitively eluted by poly peptides, and then prepared in SDS-PAGE loading buffer (pH 8.8) and analyzed immediately using western blot. The SDS-PAGE gel was freshly prepared with a modification, precisely pH 8.8 stacking gel and 10% resolving gels. Running buffer and transfer buffer were adjusted to pH 8.8. All electrophoresis steps were performed at 4 °C and samples were resolved at 80-120 V for 2–3 h. Proteins were transferred to PVDF membranes at 30 V for 15 h at 4 °C and immediately incubated for 45 min at RT in blocking buffer [5% bovine serum albumin (BSA) in TBS with 0.1% Tween-20 (0.1% TBST), pH 8.8]. 3-pHis antibody was diluted in blocking buffer, incubated with membranes for 1 h at RT. Membranes were washed three times for 10 min each with 0.1% TBST before incubation with HRP-conjugated secondary antibodies for 50 min at room temperature. Membranes were again washed three times for 10 min each with 0.1% TBST before enhanced chemiluminescence detection.

### Peptide dot blot

Peptide dot blots were used to test the specificity of PGAM1 pY119 antibody. The peptides including phosphorylated PGAM1 Y119 peptide (KIWRRSpYDVP), unphosphorylated PGAM1 Y119 peptide (KIWRRSYDVP), phosphorylated PGAM1 Y26 peptide

(RFSGWpYDADL) and phosphorylated OGT Y976 peptide (IAKNRQEpYEDIAV) were dissolved in water to reach the concentration of 2 mg/mL. In the assay, serial dilutions (200, 100, 50, 10, 5, and 1 ng/μL) were prepared and 1 μL of each peptide was spotted on nitrocellulose. After drying for 10 min at RT, membranes were blocked with blocking buffer for 1 h at RT and incubated with PGAM1 pY119 antibody (diluted 1:1000 in blocking buffer) for 1 h at RT as described in "western blot" in "Methods".

## Assay for pyruvate generation

As previously reported, the pyruvate level was measured using the Pyruvate Assay Kit (Sigma, MAK071) according to the instructions from the manufacturer (Li et al, 2023). In brief, samples from our in vitro kinase assay and standards from the kit were incubated with appropriate volumes of the reaction mix (Pyruvate Assay Buffer, Pyruvate Probe and Pyruvate Enzyme Mix) in a 96-well plate. For colorimetric assays, the absorbance at 570 nm was measured with a microplate reader. Pyruvate concentrations of the samples were calculated based on the absorbance of the standards.

## PKM2 oligomerization assay

The whole-cell lysates (200 μg/mL) or recombinant His-PKM2 proteins purified bacterially (200 μg/mL) were cross-linked with 1% formaldehyde (FA) for 20 min and terminated with SDS-PAGE loading buffer. Subsequently, samples were analyzed by western blot.

## PKM2 interactome analysis by mass spectrometry

Proteins complex (300 μg) immunoprecipitated with PKM2 antibody from A549 cells were subjected to SDS-PGAE and then in-gel protein digestion and peptides recovery. The operation steps are as follows: Cut three excised gel slices from each gel into 1-mm³ cubes, add 500 μL of 50 mM ammonium bicarbonate (NH$_4$HCO$_3$)/acetonitrile (ACN) (1:1, v/v) solution and wash until coomassie blue disappear. Remove the supernatant, add 500 μL of ACN and incubate for 10 min. Remove the ACN, rehydrate the gel slices in 10 mM DTT/50 mM NH$_4$HCO$_3$ to completely cover the gel slices, and incubate at 56 °C for 1 h. Remove the supernatant, add 500 μL of ACN and incubate for 10 min. Remove the ACN and add the 50 mM iodoacetamide (IAA)/50 mM NH$_4$HCO$_3$ to completely cover the gel slices. Incubate for 60 min at room temperature in the dark. Remove the IAA/ NH$_4$HCO$_3$, add 500 μL of ACN and incubate for 10 min. Remove the ACN solution and add just enough enzyme digestion solution to cover the gel slices. Incubate the gel pieces on ice for 45 min. Add 10 μL of enzyme digestion solution to keep the gel pieces wet during enzymatic digestion. Incubate overnight at 37 °C. Add 100 μL extraction solution (5% TFA-50% ACN-45% ddH$_2$O) at 37 °C water bath for 1 h, sonicate, centrifuge then transfer the extract to a fresh microcentrifuge tube. Lyophilize the extracted peptides to near dryness. Resuspend peptides in 10 μL of 0.1% formic acid.

The solutions containing peptides released during in-gel digestion were measured using Nanoflow UPLC (Easy-nLC 1200 system, Thermo, Cambridge, MA, USA) coupled to Mass Spectrometer (Q Exactive™ Hybrid Quadrupole-Orbitrap™,

Thermo). Briefly, the trypsinized peptides were firstly trapped and desalted on a 150 μm × 15 cm in-house made column packed with Acclaim PepMap RPLC C18 (1.9 μm, 100 Å, Dr. Maisch GmbH, Germany) with a pump system supplied moving phase A (0.1% formic acid in distilled water) and moving phase B (0.1% formic acid in acetonitrile). The profile of the gradient moving phase was as follows: 4–8% B for 2 min, from 8 to 40% B for 43 min, from 40 to 60% B for 10 min, from 60 to 95% B for 1 min and from 95 to 95% B for 10 min; and the flow rate was 600 nl/min. Different fractions of the eluate were injected into the Q-Exactive mass spectrometry set in a positive ion mode and the data-dependent manner with a full MS scan from 300 to 1800 *m/z*. High collision energy dissociation was employed as the MS/MS acquisition method. The raw MS files were analyzed and searched against protein database based on the species of the samples using MaxQuant (1.6.2.10). The parameters were set as follows: the protein modifications were carbamidomethylation (C) (fixed), oxidation (M) (variable), Acetyl (Protein N-term) (variable); the enzyme specificity was set to trypsin; the maximum missed cleavages were set to 2; the precursor ion mass tolerance was set to 20 ppm, and MS/MS tolerance was 20 ppm. Only high confident identified peptides were chosen for downstream protein identification analysis. The differentially expressed proteins identified from mass spectrometry were subjected to Kyoto Encyclopedia of Genes and Genomes (KEGG) pathways enrichment assay.

## Mass spectrometric analysis of PGAM1 phosphorylation

PGAM1 proteins from A549 cells (5 × 10⁸ cells) were isolated by IP and then subjected to SDS-PAGE. The operation steps of the in-gel protein digestion, peptides recovery and LC-MS analysis are the same as above "PKM2 interactome analysis by mass spectrometry". The raw MS files were analyzed and searched against the human PGAM1 database (UniProt_P18669) using Byonic. The parameters were set as follows: the protein modifications were carbamidomethylation (C) (fixed), oxidation (M) (variable), acetyl (Protein N-term) (variable), phosphor (S/T/Y) (variable), the enzyme specificity was set to trypsin or chymotrypsin; the maximum missed cleavages were set to 3; the precursor ion mass tolerance was set to 20 ppm, and MS/MS tolerance was 0.02 Da. Only high confident identified peptides were chosen for downstream protein identification analysis.

## PGAM1 activity assay

As previously reported, PGAM1 activity was measured with the Phosphoglycerate Mutase Activity Assay Kit (BioVision, K2007, San Francisco, CA, USA) according to instructions from the manufacturer (Hou et al, 2023). In brief, cells were lysed with cold PGAM lysis buffer and centrifuged at 10,000 × *g* at 4 °C for 10 min to remove cell debris. The supernatant samples and standards from the kit were incubated with appropriate volumes of the reaction mix (PGAM Assay Buffer, PGAM Substrate, PGAM Cofactor, PGAM Converter 1, PGAM Converter 2, PGAM Developer, and OxiRedTM Probe) in a 96-well plate. For colorimetric assays, the absorbance at 570 nm was measured with a plate reader. PGAM1 activity was calculated based on the absorbance of the standards.

## [$^{13}C_6$]-glucose tracing assessment

Cell samples were cultured in medium containing [$^{13}C_6$]-glucose (17.3 mM, Sigma) for 24 h. Polar metabolites were extracted using 1000 µl of ice-cold methanol: $H_2O$ (4:1, v/v), and incubated at 1500 rpm for 30 min at 4 °C. At the end of the incubation, samples were centrifuged for 10 min at 12,000 rpm at 4 °C. Clean supernatant was transferred to a new tube. Extracts were dried in a SpeedVac under $H_2O$ mode. The dried extract was reconstituted in 2% acetonitrile in water prior to ultra-high performance liquid chromatography-high resolution mass spectrometry (UHPLC-HRMS) analysis on an Agilent 1290 II UPLC coupled to Sciex 5600 + quadrupole-TOF MS. For reverse phase chromatography, polar metabolites were separated on a Waters ACQUITY HSS-T3 column (3.0 × 100 mm, 1.8 µm), and mobile phase A was water containing 0.1% formic acid (v/v), and mobile phase B was acetonitrile. While a Waters ACQUITY BEH Amide column (2.1 × 100 mm, 1.7 µm) was utilized for normal phase chromatographic separation, Mobile phase A was water containing 20 mM ammonium acetate in water: acetonitrile 90:10 (v/v) at pH 9, and mobile phase B was 20 mM ammonium acetate in water: acetonitrile 10:90 (v/v) at pH 9. MS parameters for detection were: ESI source voltage positive ion mode 5.5 kV, negative ion mode −4.5 kV; vaporizer temperature, 500 °C; drying gas (N2) pressure, 50 psi; nebulizer gas (N2) pressure, 50 psi; curtain gas (N2) pressure, 35 psi; The scan range was $m/z$ 60–800. Information-dependent acquisition mode was used for MS/MS analyses of the metabolites. Collision energy was set at (±) 35 ± 15 eV. Data acquisition and processing were performed using Analyst® TF 1.7.1 Software (AB Sciex, Concord, ON, Canada). All detected ions were extracted using MarkerView 1.3 (AB Sciex, Concord, ON, Canada) into Excel in the format of a two-dimensional matrix, including mass-to-charge ratio ($m/z$), retention time, and peak areas, and isotopic peaks were filtered. PeakView 2.2 (AB Sciex, Concord, ON, Canada) was applied to extract MS/MS data and perform comparisons with the Metabolites database (AB Sciex, Concord, ON, Canada), HMDB, METLIN, and standard references to annotate ion identities.

## EdU Click-iT assay

Cells were plated on glasses for 24 h and then treated with EdU (40 µM) for 20 min. Then cells were washed and fixed in 4% paraformaldehyde for 7 min. Following fixation, EdU staining was performed using Click-iT™ Cell Reaction Buffer Kit (Invitrogen, C10269) and Alexa Fluor™ 647 Azide (Invitrogen, A10277) according to the manufacturer's instructions. Cell nuclei were washed three times and stained with DAPI (Sigma, D9542). Images were acquired by fluorescent confocal microscopy (Carl Zeiss LSM880) and analyzed with software NIS-Elements AR version 5.01 (Nikon). Statistical analysis was performed using Prism 8 (GraphPad software).

## Cellular lipid-level assessment

Cell samples ($1 \times 10^7$ cells) were mixed with 200 µL of MeOH and 1 mL of MTBE, and sonicated for 10 min in ice water bath, then left for 30 min at 4 °C. Following adding 200 µL water and 50 mL internal standard into the samples and leaving for 10 min at 4 °C,

the extraction solution was centrifuged at $5000 \times g$ and 4 °C for 15 min. All the supernatants were dried and then re-dissolved in 100 µL of dichloromethane/methanol (1:1, v/v, including internal standard) to obtain samples for machine testing. An equal volume of samples was mixed from all prepared samples to obtain the quality control samples.

Chromatographic separation was performed on a ThermoFisher Ultimate 3000 UHPLC system with a Waters CSH C18 column (2.1 mm × 100 mm, 1.7 µm). For positive mode: The mobile phases consisted of (A) water and (B) isopropanol/acetonitrile (9:1, v/v), both with 10 mM ammonium formate and 0.1% formate. A linear gradient elution was performed with the following program: 0 min, 40% B; 2 min, 45% B; 4 min, 55% B and held to 10 min; 14 min, 90% B; 15 min, 95% B and held to 18 min; 18.1 min, 40% B and held to 20 min. For negative mode: The mobile phases consisted of (A) water and (B) isopropanol/acetonitrile (9:1, v/v), both with 10 mM ammonium formate. A linear gradient elution was performed with the following program: 0 min, 40% B; 2 min, 45% B; 4 min, 55% B and held to 10 min; 14 min, 90% B; 15 min, 95% B and held to 18 min; 18.1 min, 40% B and held to 20 min. The flow rate was 0.3 mL/min, and the injection volume was 2 µL both for positive mode and negative mode. The eluents were analyzed on a ThermoFisher Q Exactive™ Hybrid Quadrupole-Orbitrap™ Mass Spectrometry (QE) in Heated Electrospray Ionization Positive (HESI + ) and Negative (HESI-) mode, respectively. Spray voltage was set to 3.5 kV for HESI+ and HESI-. Both Capillary and Aux Gas Temperature were 350 °C. The sheath gas flow rate was 40 (Arb). Aux gas flow rate was 10 (Arb). S-Lens RF Level was 50 (Arb). The full scan was operated at a high resolution of 70,000 FWHM ($m/z = 200$) at a range of 130–1950 $m/z$ with AGC Target setting at $1 \times 10^6$. Simultaneously, the fragment ions information of top 10 precursors each scan was acquired by Data-dependant acquisition with HCD energy at 20, 30, and 40 eV, mass resolution of 17500 FWHM, and AGC Target of $5 \times 10^5$. The raw data of UHPLC-HRMS/MS were preprocessed by Compound Discoverer (verison 3.3, Thermo Fisher), where peak finding, peak alignment, and compound identification were performed. In addition to the default parameters, other main settings are as follows, Mass Tolerance 5 ppm, RT Tolerance 0.2 min. The in-house standard database and mzCloud database were used for compound identification, and the differential compounds are manually reviewed or re-identified. The final data was exported as a peak table file, mainly including observations (sample name), variables (rt_mz), and peak areas. The data were normalized against the internal standards before performing univariate and multivariate statistics.

## Intracellular NADPH-level assay

As previously reported, the NADPH level was measured using the NADP/NADPH quantification kit (K347, Biovision) according to the instructions from the manufacturer (Li et al, 2023). In brief, samples and standards from the kit were heated at 60 °C for 30 min to decompose NADP, and then incubated with appropriate volumes of the reaction mix (NADP/NADPH Extraction Buffer, NADP Cycling Buffer, NADP Cycling Enzyme Mix, NADPH Developer and Stop Solution) in a 96-well plate. For colorimetric assays, the absorbance at 450 nm was measured with a microplate reader. NADPH concentrations of the samples were calculated based on the absorbance of the standards.

## Intracellular ROS production

Intracellular ROS levels were quantified using the CellROX deep red flow cytometry assay kit (Invitrogen, C10422) according to instructions from the manufacturer. The cell-permeable reagents are non-fluorescent or very weakly fluorescent while in a reduced state and upon oxidation by ROS exhibit strong fluorogenic signal. The amount of intracellular ROS was measured by detecting fluorescence intensity. In brief, CellROX Deep Red Reagent was added to cells at a final concentration of 5 μM, and further incubated for 30 min at 37 °C. The cells were harvested and analyzed at excitation/emission = 640/665 nm by flow cytometry (BD, FACSCanto™II).

## Colony formation assay

For colony formation assays, $1 \times 10^4$ cells were plated in a 100-mm plate in triplicates and cultured for 15 days. The cells were washed carefully with PBS, and then fixed with 1 mL 10% methanol solution for 30 s. Following the removal of methanol, 1 mL 0.1% crystal violet dye was added for an 30 min-incubation at room temperature. Subsequently, the plate was washed with distilled water and dried at 37 °C before adding 1 mL of 33% acetic acid for decolorization and quantification. The decolorization solution was measured at OD = 595 nm. Absorbance values represent the ability of cell colony formation.

## Cell proliferation assay

In all, $5 \times 10^4$ PGAM1 WT- or Y119F mutant cells were respectively seeded in six-well plates. Cells were counted every 24 h over a 7-day period to draw up the proliferation curve.

## pY119-TAT peptide development

The cell-permeable pY119-TAT peptide, which is derived from PGAM1 and contains Y119 phosphorylation (KIWRRSpYDVP), was designed and fused with HIV-TAT (YGRKKRRQRRR): KIWRRSpYDVPYGRKKRRQRRR; MW, 2941.37. It was manufactured by Scilight Biotechnology, LLC. (Beijing, China) at greater than 95% purity and stored at −20 °C to avoid freeze-thawing artifacts. For in vitro experiments, pY119-TAT peptide was dissolved in PBS to generate a 25 mM stock. For in vivo assays, pY119-TAT peptide was dissolved in PBS to obtain a 1 mg/mL stock solution, which was kept on ice and brought to room temperature before injection.

## Tumor xenograft analysis

In total, $1 \times 10^6$ PGAM1[WT] or PGAM1[Y119F] rescued cells were resuspended with 50 μL Matrigel (Corning) and 100 μL PBS. Subsequently, the mixtures were injected into BALB/c-nude mice (nu/nu, 6–8 weeks-old females). In all, 14 days after the injection, tumors were dissected and analyzed. For the functional analysis associated with pY119-TAT peptide, $1 \times 10^6$ A549 cells were resuspended with 50 μL Matrigel and 100 μL PBS and injected into BALB/c-nude mice (nu/nu, 6–8 weeks-old females). Five days after tumor colonization, 10 mg/kg of TAT peptide or pY119-TAT peptide was injected intraperitoneally every other day. The xenograft tumors were sampled and photographed after 15 days. The nude mice were obtained from the Beijing HFK Bioscience Co., Ltd. (Beijing, China), and housed in a 12 h light and black cycle at 20 °C and 40% humidity-controlled room. All animal work procedures were approved by the Ethics Committee of Science and Technology of Northeast Normal University (202302036).

## Immunohistochemistry analysis

Mouse tumor tissues were fixed and prepared for Immunohistochemistry (IHC) as described (Yang et al, 2012b). Briefly, paraffin-cut sections of xenograft tumors were prepared, then stained with hematoxylin & eosin (HE) and Ki-67. To quantify the IHC result of positive staining, three random areas of the tissue in each sample were microscopically examined, analyzed, and imaged (Lecia DMi8).

The tissue sections from paraffin-embedded human lung cancer were stained with antibodies as indicated. We blindly scored the tissue sections according to the percentage of positive cells and staining intensity. We rated the intensity of staining on a scale of 0–3: 0 (negative), 1 (weak), 2 (moderate), and 3 (strong). We assigned the following proportion scores of 0–4: 0 (< 5%), 1 (6–25%), 2 (26–50%), 3 (51–75%) and 4 (> 75%). The IHC score was obtained by the formula: 3 × score of strongly staining area + 2 × score of moderately staining area + 1 × score of weakly staining area, giving a range of 0–12. Scores were compared with overall survival, defined as the time from the date of diagnosis to death or the last known date of follow-up.

Lung cancer tissues and matching tumor-adjacent normal tissues from patients were obtained from the Biobank of China–Japan Union Hospital of Jilin University during surgery and stored at −80 °C. The present study was approved by the Ethics Committee of Science and Technology of Northeast Normal University (202302036) and conducted with the informed consent of all patients.

## Statistical analysis

For the IHC data, Student's *t* test (paired, two-tailed) was used for two group comparisons (Fig. 6B; Appendix Fig. S4A,B). Pearson's correlation test was used to analyze the statistical significance of the correlation of IHC staining between proteins in the human lung cancer tissue data (Fig. 6D,E). Log-rank test was used to indicate the statistical significance of survival correlation between groups (Fig. 6F). Other than IHC data, Student's *t* tests (unpaired) were performed for data with two groups. A one-way or two-way analysis of variance (ANOVA) was performed followed by Tukey's multiple comparison test for datasets with three or more groups. Data were analyzed using Prism Version 8 software (GraphPad) and presented as the mean ± SD values. ****$P < 0.0001$, ***$P < 0.001$, **$P < 0.01$, *$P < 0.05$, ns nonsignificant.

# Data availability

All data have been included in the manuscript, figures, extended view information, appendix figures and the Source Data files. The mass spectrometry proteomics data of PGAM1 phosphorylation have been deposited to the ProteomeXchange Consortium via the PRIDE (Perez-Riverol et al, 2022) partner repository with the

dataset identifier PXD049235 (https://www.ebi.ac.uk/pride/archive/projects/PXD049235). The mass spectrometry lipidomics data of cellular lipid levels are available via the PRIDE partner repository with the dataset identifier PXD049310 (https://www.ebi.ac.uk/pride/archive/projects/PXD049310).

The source data of this paper are collected in the following database record: biostudies:S-SCDT-10_1038-S44318-024-00110-8.

## Peer review information

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

## Acknowledgements

The authors thank Dr. Weiwei Yang (Shanghai Institute of Biochemistry and Cell Biology, Chinese Academy of Sciences) for providing the Src plasmid. The authors thank the Biobank of China–Japan Union Hospital of Jilin University for providing biospecimens and data. This work was supported by the National Natural Science Foundation of China (32101048, 32070896, 32070758), the Young Scientific and Technological Talents Support Project of Jilin Province (QT202105) and the China Postdoctoral Science Foundation (2020T130090, 2019M661190).

## Author contributions

**Yang Wang**: Conceptualization; Data curation; Formal analysis; Funding acquisition; Investigation; Methodology; Writing—original draft; Project administration; Writing—review and editing. **Hengyao Shu**: Conceptualization; Software; Investigation; Methodology. **Yanzhao Qu**: Investigation. **Xin Jin**: Software; Visualization; Methodology. **Jia Liu**: Conceptualization; Data curation; Methodology. **Wanting Peng**: Investigation; Methodology. **Lihua Wang**: Data curation; Software; Investigation. **Miao Hao**: Resources. **Mingjie Xia**: Resources. **Zhexuan Zhao**: Software. **Kejian Dong**: Software. **Yao Di**: Investigation. **Miaomiao Tian**: Conceptualization. **Fengqi Hao**: Conceptualization. **Chaoyi Xia**: Conceptualization. **Wenxia Zhang**: Conceptualization; Writing—original draft; Project administration; Writing—review and editing. **Yunpeng Feng**: Conceptualization; Funding acquisition; Writing—original draft; Project administration; Writing—review and editing. **Min Wei**: Conceptualization; Funding acquisition; Writing—original draft; Writing—review and editing.

Source data underlying figure panels in this paper may have individual authorship assigned. Where available, figure panel/source data authorship is listed in the following database record: biostudies:S-SCDT-10_1038-S44318-024-00110-8.

## Disclosure and competing interests statement

The authors declare no competing interests.

# Expanded View Figures

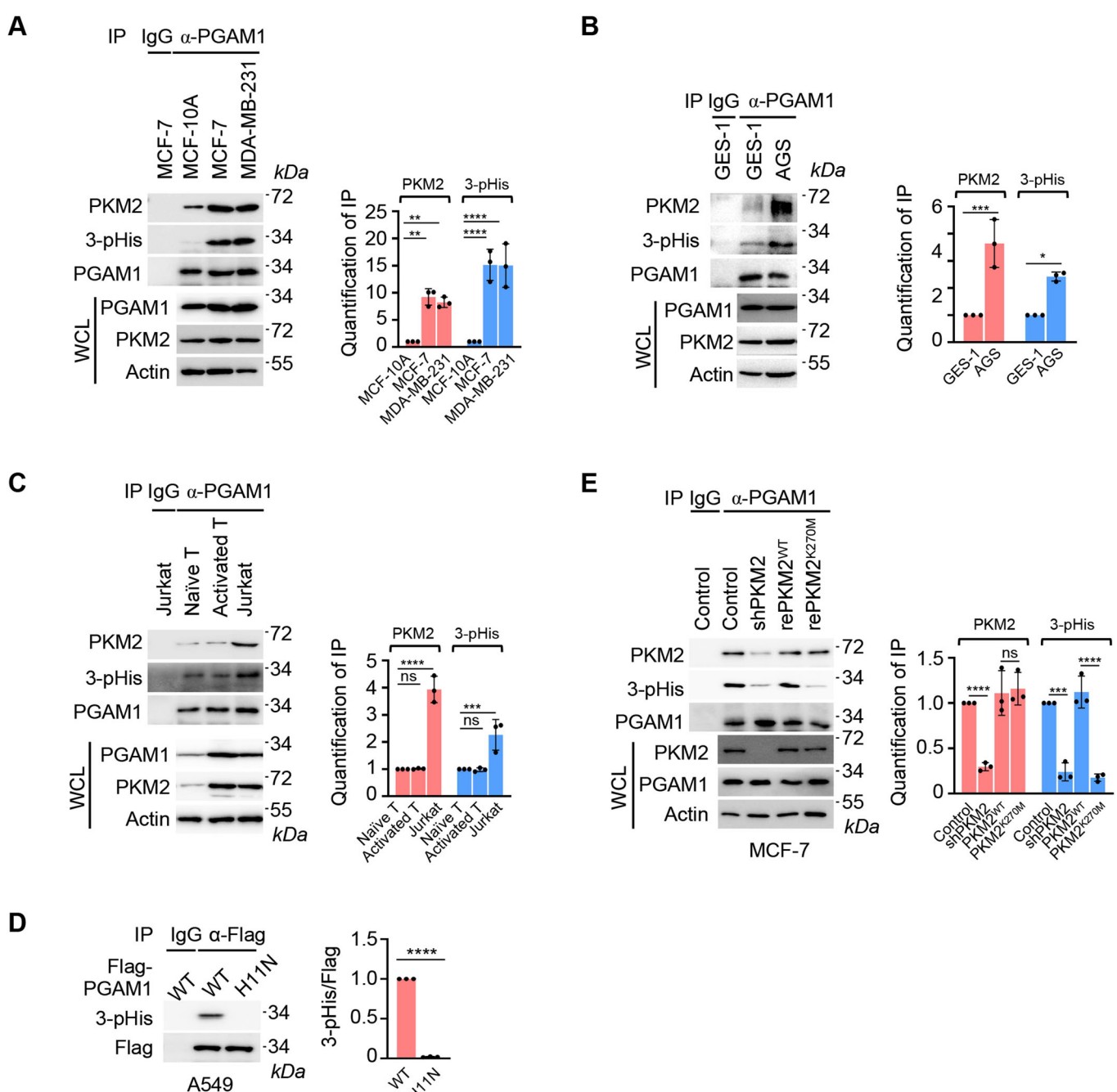

**Figure EV1. PKM2 upregulates PGAM H11 phosphorylation.**

(A–C) PGAM1–PKM2 interaction and PGAM1 phosphorylation are more evident in tumor cells. Left: PGAM1-associated proteins were immunoprecipitated from the whole-cell lysates of different tumor cells and analyzed by WB. Right: Quantification of IP. Relative levels of PKM2 or 3-pHis were normalized to that of PGAM1 for each group. (D) PGAM1 H11N mutation blocks 3-pHis modification. A549 cells were transfected with Flag- PGAM1 WT or H11N mutation for 48 h. Left: Flag (PGAM1)-associated proteins were immunoprecipitated and analyzed by WB. Right: Quantification of IP. Relative levels of 3-pHis were normalized to that of Flag (PGAM1) for each group. (E) PKM2 K270M mutation reduces PGAM1 H11 phosphorylation. PKM2-depleted MCF-7 cells were re-expressed PKM2 WT or PKM2 K270M mutation. Left: PGAM1-associated proteins were immunoprecipitated and analyzed by WB. Right: Quantification of IP. Relative levels of PKM2 or 3-pHis were normalized to that of PGAM1 for each group. Data information: for WB in (A–E), one representative experiment out of three was shown. IgG served as a negative control. Data represent mean ± SD of three (A–E) independent experiments, with significance determined by one-way ANOVA test (A–C, E) or Student's $t$ test (D); $****P < 0.0001$, $***P < 0.001$, $**P < 0.01$, $*P < 0.05$, ns, nonsignificant. Source data are available online for this figure.

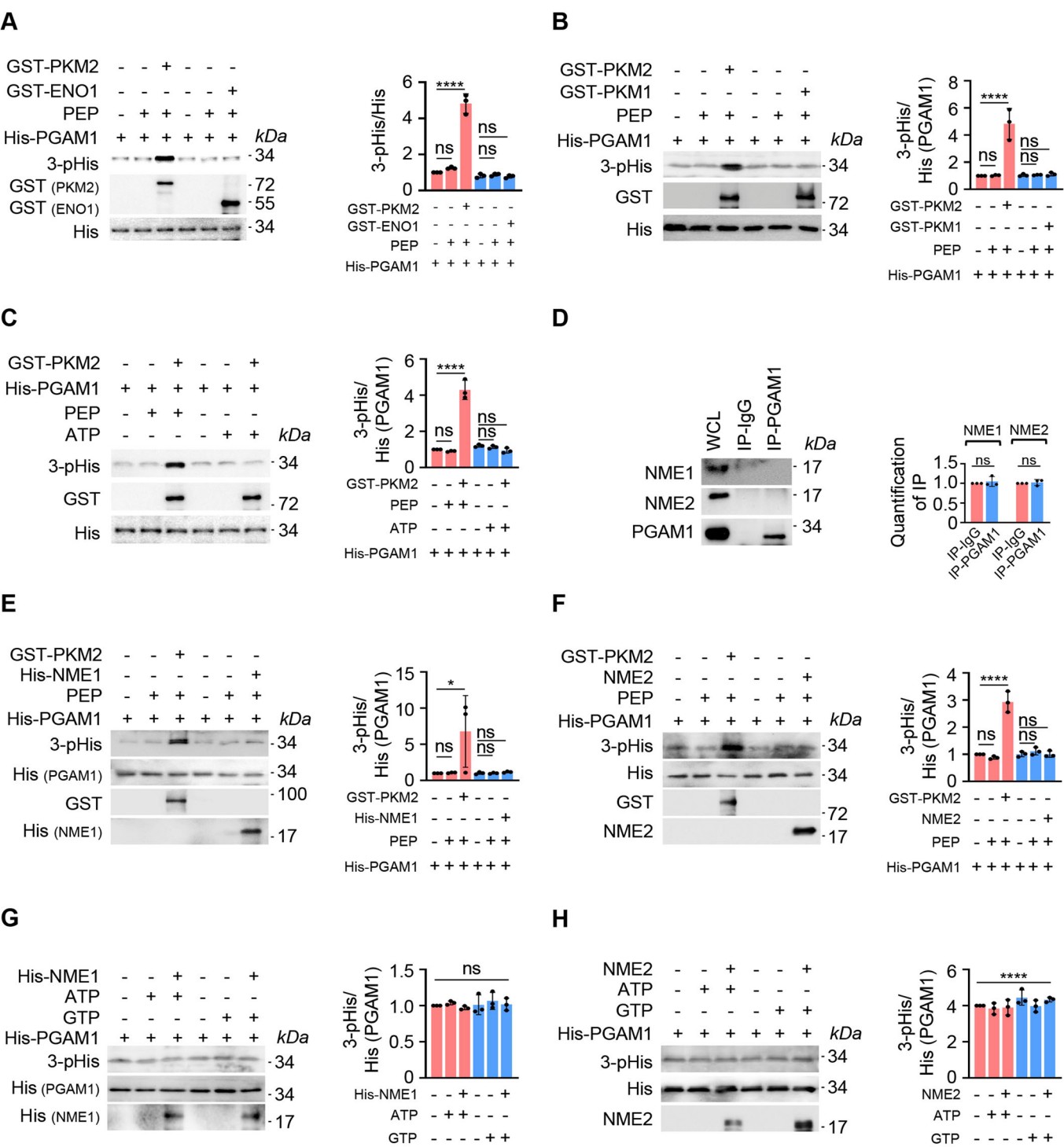

**◀**

**Figure EV2. ENO1, PKM1, ATP, NME1 or NME2 does not impact PGAM1 H11 phosphorylation.**

(A–C) ENO1, PKM1 or ATP does not affect PGAM1 H11 phosphorylation. Left: In vitro kinase assay was carried out with recombinant GST-ENO1, GST-PKM2, GST-PKM1 and His-PGAM1 in the presence of PEP or ATP. PGAM1 H11 phosphorylation was detected by WB. Right: Quantification of WB. Relative levels of 3-pHis were normalized to that of His (PGAM1) for each group. (D) PGAM1 does not interact with NME1 or NME2. Left: PGAM1-associated proteins in A549 cells were immunoprecipitated and analyzed by WB. IgG served as a negative control. Right: Quantification of IP. Relative levels of NME1 or NME2 in IP were normalized to that in WCL for each group. (E–H) NME1 or NME2 does not affect PGAM1 H11 phosphorylation. Left: In vitro kinase assay was carried out with recombinant GST-PKM2, His-NME1, NME2 and His-PGAM1 in the presence of PEP, ATP or GTP. PGAM1 H11 phosphorylation was detected by WB. Right: Quantification of WB. Relative levels of 3-pHis were normalized to that of His (PGAM1) for each group. Data information: for WB in (A–H), one representative experiment out of three was shown. Data were represented as mean ± SD of three independent experiments (A–H) with significance determined by one-way ANOVA test (A–C, E–H) and Student's *t* test (D); ****$P < 0.0001$, *$P < 0.05$, ns nonsignificant. Source data are available online for this figure.

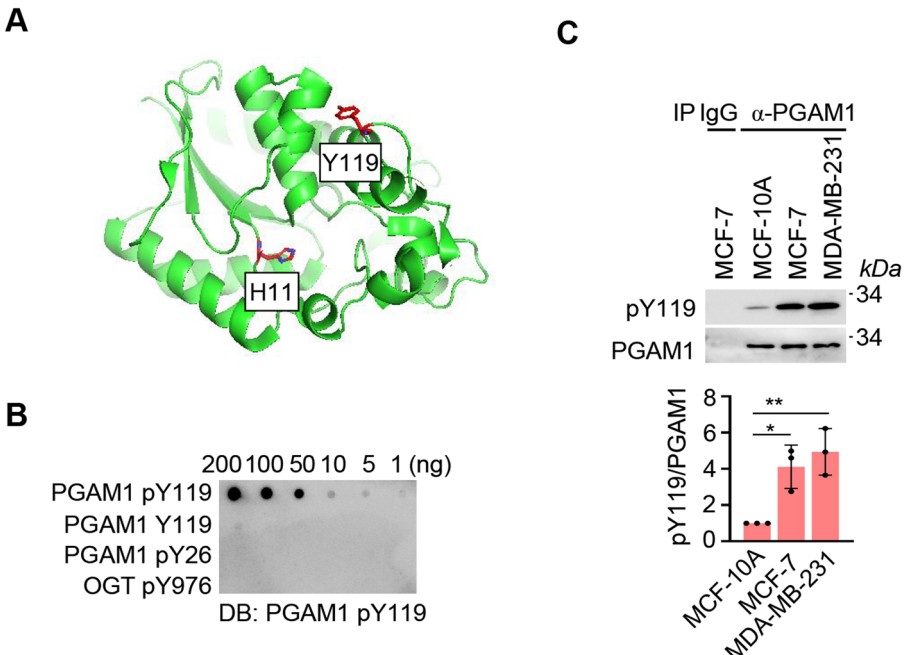

**Figure EV3.   PGAM1 Y119 molecular location and Y119 phosphorylation analysis in different cells.**

(A) Molecular location of Y119 and H11 in PGAM1. A monomer of PGAM1 from protein data bank (PDB, ID: 4GPI) was shown in green cartoon. Y119 and H11 is shown in red sticks. (B) Confirmation of PGAM1 pY119 antibody by dot blot (DB). Titrated amounts of peptides, including phosphorylated PGAM1 Y119, unphosphorylated PGAM1 Y119, phosphorylated PGAM1 Y26 and phosphorylated OGT Y976, were spotted on nitrocellulose membrane and then probed with PGAM1 pY119 antibody. (C) PGAM1 Y119 phosphorylation is more evident in tumor cells. Top: In MCF-10A, MCF-7 and MDA-MB-231 cell lines, PGAM1-associated proteins were immunoprecipitated and analyzed by WB. One representative experiment out of three was shown. IgG served as a negative control. Bottom: Quantification of IP. Relative levels of pY119 were normalized to that of PGAM1 for each group. Data represents mean ± SD ($n = 3$) with significance determined by one-way ANOVA test; $^{**}P < 0.01$, $^{*}P < 0.05$. Source data are available online for this figure.

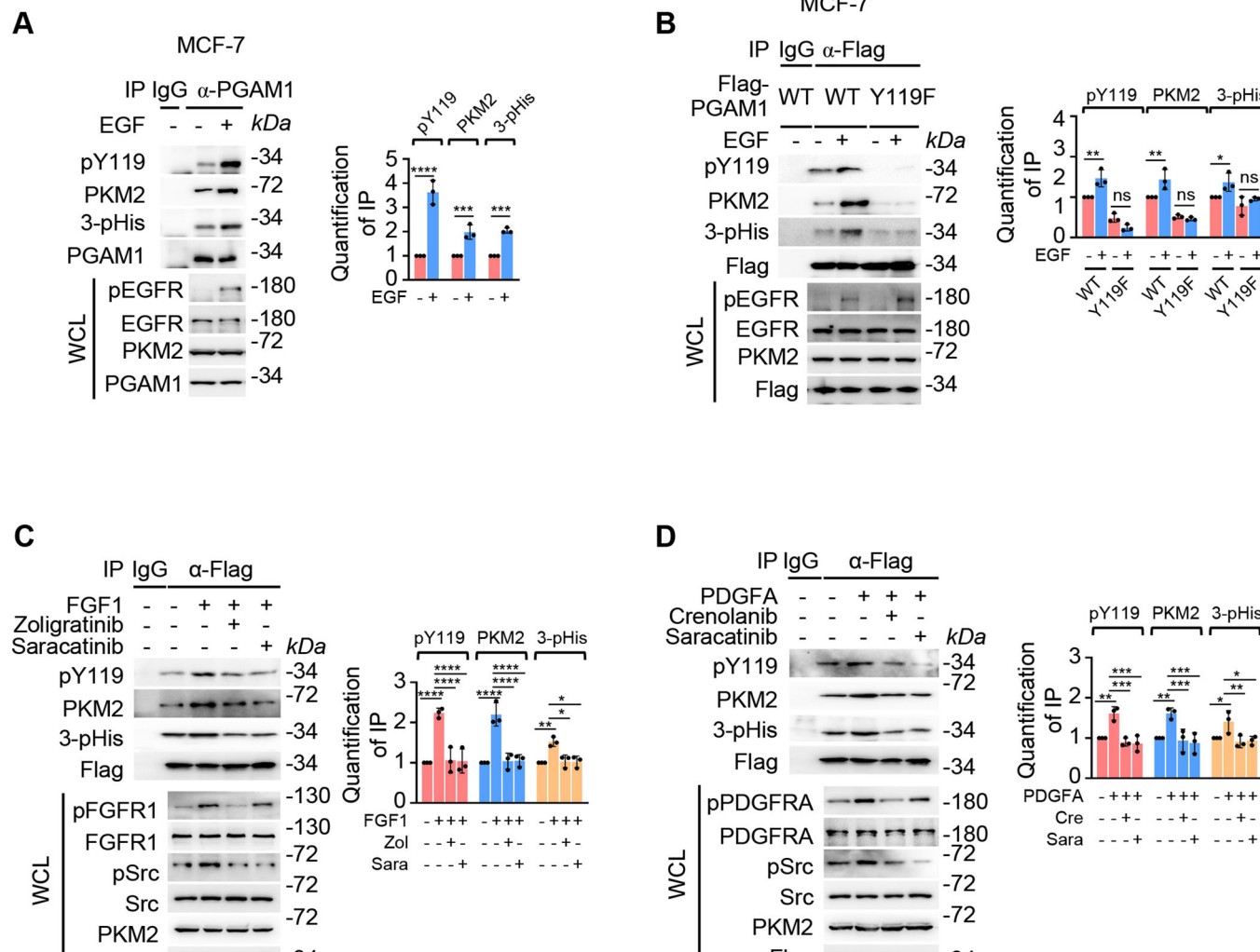

**Figure EV4.  Growth factors' signaling regulates PGAM1 Y119 phosphorylation.**

(**A, B**) EGF triggers PGAM1 Y119 phosphorylation. Left: MCF-7 cells, or MCF-7 cells transfected with Flag-PGAM1 plasmids were cultured in the serum-free medium for 48 h, and then treated with or without EGF (100 ng/ml) for 30 min. Flag (PGAM1)-associated proteins were immunoprecipitated and analyzed by WB. Right: Quantification of IP. Relative levels of pY119, PKM2 or 3-pHis were normalized to that of endogenous or exogenous PGAM1 for each group. (**C, D**) FGF and PDGF trigger PGAM1 Y119 phosphorylation. Left: A549 cells were transfected with Flag-PGAM1 for 6 h, then cultured in serum-free medium and simultaneously treated with Zoligratinib (10 μM), Crenolanib (20 pM), Saracatinib (1 μM) for 48 h. Subsequently, the cells were incubated with FGF1 and PDGFA (50 ng/ml) for 6 h. Flag (PGAM1)-associated proteins were immunoprecipitated and analyzed by WB. Right: Quantification of IP. Relative levels of pY119, PKM2 or 3-pHis were normalized to that of Flag (PGAM1) for each group. Data information: for WB in (**A–D**), one representative experiment out of three was shown. IgG served as a negative control. Data represent mean ± SD of three (**A–D**) independent experiments with significance determined by two-way ANOVA test; ****$P < 0.0001$, ***$P < 0.001$, **$P < 0.01$, *$P < 0.05$. ns nonsignificant. Source data are available online for this figure.

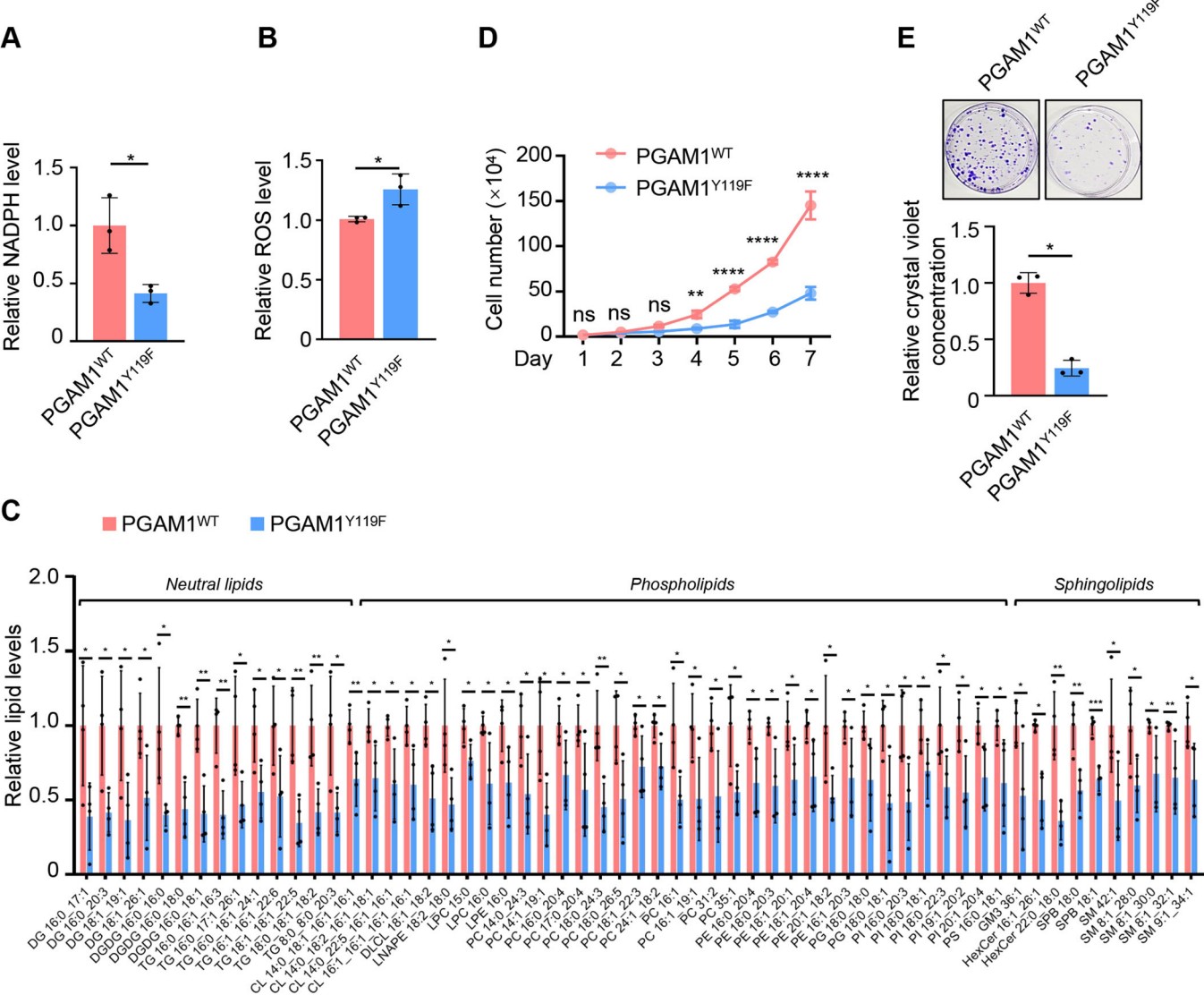

**Figure EV5. PGAM1 Y119 mutation compromises cell proliferation.**

(A, B) PGAM1 Y119 mutation impairs redox balance. Intracellular NADPH (A) and ROS (B) levels in PGAM1 WT- and Y119F mutation- rescued cells were examined with commercially available kits. $n = 3$ independent experiments. (C) Impact of PGAM1 Y119 mutation on lipid levels. Lipid levels in PGAM1 WT- and Y119F mutant-restored cells were analyzed by UHPLC-HRMS, which is related to Fig. 5D. $n = 4$ biological replicas (D) PGAM1 Y119 mutation hampers cell proliferation. PGAM1 WT- and Y119F mutant-restored cells were seeded with the same number and then counted every 24 h. The proliferation curves were drawn. $n = 3$ independent experiments. (E) PGAM1 Y119 mutation inhibits cell colony formation. PGAM1 WT- and Y119F mutation-rescued cells were seeded at the same number and cultured for 20 days. Cell colonies were stained by crystal violet. Top: Representative images and (bottom) quantifications were shown. $n = 3$ independent experiments. Data information: data represents mean ± SD ($n \geq 3$) with significance determined by Student's $t$ test (A–E); ****$P < 0.0001$, ***$P < 0.001$, **$P < 0.01$, *$P < 0.05$. ns nonsignificant. Source data are available online for this figure.

