## [Peer Review File · The EMBO Journal]

PKM2 functions as a histidine kinase to phosphorylate PGAM1 and increase glycolysis shunts in cancer

Min Wei, Yang Wang, Hengyao Shu, Yanzhao Qu, Xin Jin, Jia Liu, Wanting Peng, Lihua Wang, Miao Hao, Mingjie Xia, Zhexuan Zhao, Kejian Dong, Yao Di, Miaomiao Tian, Fengqi Hao, Chaoyi Xia, Wenxia Zhang, Xueqing Ba, and Yunpeng Feng

Corresponding authors: Min Wei (weim750@nenu.edu.cn) , Xueqing Ba (baxq755@nenu.edu.cn), Yunpeng Feng (fengyp0108@nenu.edu.cn)

Review Timeline:

Submission Date:	28th Sep 23
Editorial Decision:	17th Nov 23
Revision Received:	10th Feb 24
Editorial Decision:	19th Mar 24
Revision Received:	24th Mar 24
Accepted:	27th Mar 24

Editor: Daniel Klimmeck

Transaction Report:

Dear Dr. Wei,

Thank you again for submitting your manuscript EMBOJ-2023-115733 for consideration by the EMBO Journal. Please accept my sincere apologies for getting back to you with unusual protraction due to delayed referee input, as well as detailed discussion in the editorial team. As indicated, your manuscript has been seen by three referees with expertise in cancer biology and cellular metabolism, and we have received reports from all of them, which are shown below.

Given the referees' positive recommendations, I would like to invite you to submit a revised version of the manuscript, addressing the comments of all three reviewers. I should add that it is EMBO Journal policy to allow only a single round of revision, and acceptance of your manuscript will therefore depend on the completeness of your responses in this revised version.

I would appreciate if you could contact me during the next weeks for exchange e.g. via a video call to discuss your perspective on the comments and potential plan for revisions.

Please feel free to contact me if you have any questions or need further input on the referee comments.

When submitting your revised manuscript, please carefully review the instructions below.

Please feel free to approach me any time should you have additional questions related to this.

Thank you for the opportunity to consider your work for publication.

I look forward to your revision.

Kind regards,

Daniel Klimmeck

Daniel Klimmeck, PhD
Senior Editor
The EMBO Journal

Instruction for the preparation of your revised manuscript:

2) individual production quality figure files as .eps, .tif, .jpg (one file per figure).

3) a .docx formatted letter INCLUDING the reviewers' reports and your detailed point-by-point response to their comments. As part of the EMBO Press transparent editorial process, the point-by-point response is part of the Review Process File (RPF), which will be published alongside your paper.

4) a complete author checklist, which you can download from our author guidelines ([https://wol-prod-cdn.literatumonline.com/pb-assets/embo-site/Author Checklist%20-%20EMBO%20J-1561436015657.xlsx](https://wol-prod-cdn.literatumonline.com/pb-assets/embo-site/Author%20Checklist%20-%20EMBO%20J-1561436015657.xlsx)). Please insert information in the checklist that is also reflected in the manuscript. The completed author checklist will also be part of the RPF.

6) It is mandatory to include a 'Data Availability' section after the Materials and Methods. Before submitting your revision, primary datasets produced in this study need to be deposited in an appropriate public database, and the accession numbers and database listed under 'Data Availability'. Please remember to provide a reviewer password if the datasets are not yet public (see <https://www.embopress.org/page/journal/14602075/authorguide#datadeposition>).

7) Our journal encourages inclusion of *data citations in the reference list* to directly cite datasets that were re-used and obtained from public databases. Data citations in the article text are distinct from normal bibliographical citations and should directly link to the database records from which the data can be accessed. In the main text, data citations are formatted as follows: "Data ref: Smith et al, 2001" or "Data ref: NCBI Sequence Read Archive PRJNA342805, 2017". In the Reference list, data citations must be labelled with "[DATASET]". A data reference must provide the database name, accession number/identifiers and a resolvable link to the landing page from which the data can be accessed at the end of the reference. Further instructions are available at .

8) At EMBO Press we ask authors to provide source data for the main and EV figures. Our source data coordinator will contact you to discuss which figure panels we would need source data for and will also provide you with helpful tips on how to upload and organize the files.

Numerical data can be provided as individual .xls or .csv files (including a tab describing the data). For 'blots' or microscopy, uncropped images should be submitted (using a zip archive or a single pdf per main figure if multiple images need to be supplied for one panel). Additional information on source data and instruction on how to label the files are available at .

9) We replaced Supplementary Information with Expanded View (EV) Figures and Tables that are collapsible/expandable online (see examples in <https://www.embopress.org/doi/10.15252/emj.201695874>). A maximum of 5 EV Figures can be typeset. EV Figures should be cited as 'Figure EV1, Figure EV2' etc. in the text and their respective legends should be included in the main text after the legends of regular figures.

11) For data quantification: please specify the name of the statistical test used to generate error bars and P values, the number (n) of independent experiments (specify technical or biological replicates) underlying each data point and the test used to calculate p-values in each figure legend. The figure legends should contain a basic description of n, P and the test applied. Graphs must include a description of the bars and the error bars (s.d., s.e.m.).

We realize that it is difficult to revise to a specific deadline. In the interest of protecting the conceptual advance provided by the work, we recommend a revision within 3 months (15th Feb 2024). Please discuss the revision progress ahead of this time with

the editor if you require more time to complete the revisions.

Referee #1:

In "PKM2 functions as a histidine kinase to phosphorylate PGAM1 to increase glycolysis shunts in cancer", Wang, Shu, and colleagues explore a novel function of PKM2 in phosphorylating and modulating PGAM1 in cancer to change the fate of glucose in glycolysis. They first demonstrate that PKM2 as the kinase and PEP as the substrate are responsible for PGAM1 H11 phosphorylation in cancer cells, and that this does not occur as much in non-cancer cells. Their data suggest that dimers and monomers of PKM2 promote PGAM1 phosphorylation while PKM2 tetramers seem to block this process. They also use deep molecular characterization to show that Src-mediated phosphorylation, downstream of EGF, of PGAM1 Y119 is required to prime the protein for further H11 phosphorylation. Phosphorylation of PGAM1 rewires glycolysis to favor more pentose phosphate pathway flux and biosynthesis, both important for cancer cell proliferation and survival. Along these lines, the Authors show that preventing PGAM1 from being phosphorylated, either through mutations or a blocking peptide, slows tumor growth in mice. Finally, PGAM1 phosphorylation in human tumors correlates with poorer prognosis. The Authors conclude that PGAM1 phosphorylation by Src and PKM2 represents a critical node in altered cancer cell metabolism that can be exploited in therapy.

Overall, this is a well-performed study, with deep molecular analysis and well-chosen controls. I am also enthusiastic about the potential connections to cancer proliferation, and the correlations shown in human cancer. I have some requests for major revisions below that do not go beyond the scope of the study, but instead ensure that each experiment has proper controls and is properly validated. I also request that the Authors explain what cancer cell models they have chosen, and why. I have also made some more minor suggestions on writing. I expect that once published, this manuscript will have a large impact in the field, and look forward to reviewing a revised version.

Major comments

- I understand that this Manuscript is supposed to be a pan-cancer analysis of different cancer cell types, but the Authors must spend some space at the start of the Results section explaining what models they chose for cancer cells and non-cancer cells, and why they chose them. They must also explain why A549 were chosen for the molecular and tumor studies in the rest of the Manuscript, among all the cell types they initially used.
- In Figure 2C and D, the data on the K220R mutant do not match the Author's conclusions. The data show that K220R has high levels of both the tetramer and the dimer of PKM2, but the Authors conclude that the dimer and monomer promote PGAM1 H11 phosphorylation. In the K220R setting, there is ample monomer and dimer, but H11 phosphorylation is still blocked. Would this not instead suggest that rather than the monomer and dimer promoting H11 phosphorylation, the tetramer blocks phosphorylation? This should be addressed in the text of the Results and Discussion section.
- In Figure 4, the Authors must show in a separate experiment that each kinase inhibitor is on target at the dose the Authors selected by showing that the drug abrogates phosphorylation of a target of the kinase.
- It is not clear why some western blots have quantitations and others do not. All western blots should be quantified from replicates, and statistical analysis performed, as is done in Figure 4.
- In Figure 5B, the metabolomics heat map is missing statistical analysis to show which of the metabolites were significantly different between cells with PGAM1 WT and PGAM1 Y119F.
- On Results Section 5, the Authors state that "Because of the critical role of PPP and SSP in nucleotides synthesis and NADPH production, we further investigated the impact of PGAM1 Y119F mutation on these processes, and the result showed the impairment resulted from the mutation as expected (Fig. 5C and S7A)." However, Figure 5C is an Edu incorporation assay (which measures S phase in cell cycle), not a nucleotide synthesis assay. Is the nucleotide synthesis assay missing? Either include the missing assay and take out the Edu assay, or properly explain Figure 5c in the text as a proliferation assay.
- For Figure 6A: Since the PGAM1 pY119 antibody is a new antibody the Authors made, they need to show it is on-target and detecting Y119 in IHC by using a knockout or mutation model. The tumors from Figure 5F-G would work nicely this purpose.
- For Figure 6I, the Authors should show that the pY119-TAT peptide affects the glycolysis pathway in the tumors, for instance by looking at PGAM1 pH11, or some other direct marker.

Minor comments

- In the title for Results Section 3, the last word should be "recognition", "recognition" is not an English word.
- In Results Section 3, "Intriguingly, PGAM1 is...", the correct word is "conserved", not "conversed".
- In Figure 3F, the non-mammal species are not "lower" than mammals, this is scientifically incorrect. Please remove "lower" from the text and Figure.
- At the bottom of Results Section 3, the Authors state, "In addition, we also noticed that PGAM1WT and PGAM1Y119F no longer had different impact on PGAM1 H11 phosphorylation if PKM2 was depleted (Fig. 3I), further confirming that Y119-mutation-caused influence was inflicted by PKM2." I do not understand this sentence. I know the Authors do not conclude that PKM2 phosphorylates PGAM1 on Y119 since they address this in the next Figure, but this sentence, as written, suggests that.

Please rewrite this sentence to more clearly state the conclusions.

Referee #2:

Increased expression/activation of the glycolytic enzyme phosphoglycerate mutase 1 (PGAM1) is a common event across a wide variety of cancers. A previous study from Vander Heiden et al. revealed that PGAM1 is activated by phosphorylation at the catalytic histidine residue (His11) with phosphoenolpyruvate serving as the phosphate donor (PMID: 20847263). The nature of the enzyme that catalyses phosphate transfer from PEP to PGAM1 was not identified by Vander Heiden and colleagues. In the current study, Wang et al. provide evidence that pyruvate kinase M2 (PKM2) acts as a histidine kinase to catalyse PGAM1 H11 phosphorylation. In addition, the authors demonstrate that phosphorylation of PGAM1 at Tyr119 is required for PKM2 binding and subsequent His11 phosphorylation. Finally, the authors provide evidence that a PGAM1-derived phospho-Tyr119 peptide or mutation of Tyr119 inhibits the interaction of PGAM1 with PKM2 to disrupt glycolytic metabolism and inhibit tumour growth. The conclusions in this paper are supported by robust and compelling experimental data. Given that this study reveals important insights into the regulation of glycolytic metabolism the findings should be of interest to researchers spanning multiple fields of research. My only comments/suggestions are minor and are outlined below.

With respect to Figure 4C/D, it would be good to include a readout of the activation state of the EGFR signalling pathway (e.g. ERK phosphorylation).

On page 10, the authors state "NADPH is not only a classic antioxidant that quenches ROS...". NADPH is not antioxidant. Instead, NADPH provides reducing equivalents to generate reduced forms of antioxidant molecules. This statement should be corrected.

In Figure 5F, WT A549 xenografts grew to an average size of approximately 600 mm³ after 14 days. In Figure 6I, WT A549 xenografts grew to only approximately 200 mm³ after 14 days. While differences in tumour growth kinetics between experiments are not unusual, why were the experiments in Figure 6I stopped when tumours were so small?

With respect to Figure 6I, animal bodyweight over the course of the experimental period should be included as supplementary data.

With respect to Figure 6I, pY119-TAT peptide was administered from the time of tumour cell injection. It would be more impactful to demonstrate the impact of pY119-TAT peptide on tumour growth once tumours are already established.

The manuscript should be carefully edited to correct grammatical errors, spelling mistakes and issues with sentence structure.

Referee #3:

The manuscript by Wang et al reported identification of PKM2 as a "moonshine" histidine kinase in a phosphoenolpyruvate (PEP)-dependent manner to catalyze PGAM1 H11 phosphorylation. The authors also showed that stimulation by EGF activates Src, which in turn phosphorylates PGAM1 at Y119. The Y119 phosphorylation contributes to recruitment of PKM2 to PAGM1 and achieving H11 phosphorylation. Lastly, the authors claimed that disruption of PGMA1-PKM2 binding by a PGAM1-derived pY119-containing cell permeable peptide or Y119 mutation on PGAM1 reduces H11 phosphorylation of PGAM1, leading to reduced tumor growth.

Overall the study is well conducted and well controlled, and the findings are of some interests with mechanistic insights. This reviewer only has a couple of concerns that need to be addressed by additional experimental evidence:

1. It is well known that in mammalian cells, the cofactor 2, 3-BPG of PGAM1 is responsible for phosphorylation of H11 by transferring its phosphate group at C3 position to H11. Matt Vander Heiden reported in 2010 that 2,3-BPG levels were increased in PKM2-expressed cells as compared to PKM1-expressed cells (Vander Heiden, 2010, Science, 329, 1492-1499). Thus it is possible that in cells, PKM2 may increase 2,3-BPG levels for phosphorylation of PGAM1 at H11. The authors need to convincingly exclude this possibility before draw the conclusion that PKM2 phosphorylates PGAM1 directly and using PEP as the phosphate providing, because all of the studies performed are in vitro that cannot fully represent the physiological situation.
2. The authors need to convincingly demonstrate whether endogenous PKM2 and PGAM1 interact in diverse tumor cells, and whether such binding is regulated by EGF stimulation. Moreover, PGAM1 was reported to be phosphorylated at Y119 in cells expressing oncogenic FGFR1 (Hitosugi, 2013, Nature Communications). Is it all depending on Src? Do diverse oncogenic tyrosine kinases including FGFR1, FGFR3, ALK, PDGFRA, PDGFRB etc directly phosphorylate PGAM1 at Y119 or they have to activate Src for Y119 phosphorylation?

Point-by-point letter**Reviewers' comments:****Referee #1:**

In "PKM2 functions as a histidine kinase to phosphorylate PGAM1 to increase glycolysis shunts in cancer", Wang, Shu, and colleagues explore a novel function of PKM2 in phosphorylating and modulating PGAM1 in cancer to change the fate of glucose in glycolysis. They first demonstrate that PKM2 as the kinase and PEP as the substrate are responsible for PGAM1 H11 phosphorylation in cancer cells, and that this does not occur as much in non-cancer cells. Their data suggest that dimers and monomers of PKM2 promote PGAM1 phosphorylation while PKM2 tetramers seem to block this process. They also use deep molecular characterization to show that Src-mediated phosphorylation, downstream of EGF, of PGAM1 Y119 is required to prime the protein for further H11 phosphorylation. Phosphorylation of PGAM1 rewires glycolysis to favor more pentose phosphate pathway flux and biosynthesis, both important for cancer cell proliferation and survival. Along these lines, the Authors show that preventing PGAM1 from being phosphorylated, either through mutations or a blocking peptide, slows tumor growth in mice. Finally, PGAM1 phosphorylation in human tumors correlates with poorer prognosis. The Authors conclude that PGAM1 phosphorylation by Src and PKM2 represents a critical node in altered cancer cell metabolism that can be exploited in therapy.

Overall, this is a well-performed study, with deep molecular analysis and well-chosen controls. I am also enthusiastic about the potential connections to cancer proliferation, and the correlations shown in human cancer. I have some requests for major revisions below that do not go beyond the scope of the study, but instead ensure that each experiment has proper controls and is properly validated. I also request that the Authors explain what cancer cell models they have chosen, and why. I have also made some more minor suggestions on writing. I expect that once published, this manuscript will have a large impact in the field, and look forward to reviewing a revised version.

Response: We appreciate the reviewer's recognition of the quality of this work, as well as his/her enthusiasm about the potential links of this study with the clinical application. We are pleased to perform a number of new experiments, and provide additional evidence to strengthen the mechanism we are proposing.

Major comments

- I understand that this Manuscript is supposed to be a pan-cancer analysis of different cancer cell types, but the Authors must spend some space at the start of the Results section explaining what models they chose for cancer cells and non-cancer cells, and why they chose them. They must also explain why A549 were chosen for the molecular and tumor studies in the rest of the Manuscript, among all the cell types they initially used.

Response: Thank you for the suggestion regarding the description of the cell line selection in

the present study. Because the link between PEP and PGAM1 H11 phosphorylation was firstly observed in human non-small cell lung cancer (NSCLC) cells¹, we decided to use NSCLC A549 cell line as a representative to investigate the significance and mechanism of PKM2-PGAM1 interaction in most experiments of the present study. We made space for these interpretations, which were integrated in the revised manuscript and highlighted, shown as below for the convenience of the reviewer:

Line 92-98, Page 4-5:

PKM2-PGAM1 interaction was further confirmed in a panel of highly proliferating tumor cells, including A549, H1299, MCF-7, MDA-MB-231, AGS and Jurkat, but hardly detectable in untransformed cells, including HBE-2, MCF-10A, GES-1, activated T and naïve T cells (Fig. 1F and EV2A-C), suggesting a pan-cancer existence of the interaction between these two metabolic enzymes. Since human non-small cell lung cancer (NSCLC) was first documented for PEP based-PGAM1 phosphorylation¹, A549 cell line was chosen as a representative to investigate the significance and mechanism of PKM2-PGAM1 interaction in most experiments of the present study.

• In Figure 2C and D, the data on the K220R mutant do not match the Author's conclusions. The data show that K220R has high levels of both the tetramer and the dimer of PKM2, but the Authors conclude that the dimer and monomer promote PGAM1 H11 phosphorylation. In the K220R setting, there is ample monomer and dimer, but H11 phosphorylation is still blocked. Would this not instead suggest that rather than the monomer and dimer promoting H11 phosphorylation, the tetramer blocks phosphorylation? This should be addressed in the text of the Results and Discussion section.

Response: We fully agree with the reviewer that “PKM2 tetramerization blocks PGAM1 phosphorylation” would be more accurate. It is quite reasonable that tetramerization of PKM2 buries the appropriate interface accommodated for the substrate proteins' docking. This was addressed in the Abstract, Results and Discussion sections of the revised manuscript and highlighted, shown as below:

Line 19-22, Page 2 (Abstract):

Here, we unveil that pyruvate kinase M2 (PKM2) moonlights as a histidine kinase in a phosphoenolpyruvate (PEP)-dependent manner to catalyze PGAM1 H11 phosphorylation, that is essential for PGAM1 activity. Moreover, monomeric and dimeric PKM2 are efficient to phosphorylate and activate PGAM1, while the tetrameric PKM2 not.

Line 158-167, Page 7 (Results Section 2):

Furthermore, different PKM2 mutants, including K422R (tending to form tetramer), R399E (tending to form dimer) and Y105E (tending to form monomer)² were constructed and over-expressed in the cells lacking endogenous PKM2 (Fig. 2C). The result showed that the ratio of PKM2 tetramer significantly increased due to K422R substitution, which led to impaired PKM2-PGAM1 interaction and PGAM1 H11 phosphorylation; whereas, overexpression of PKM2 mutants bearing R399E or Y105E resulted in an enhanced PKM2-PGAM1 interaction along with PGAM1 H11 phosphorylation (Fig. 2C). In addition, the impacts of K422R, R399E and Y105E mutations on PGAM1 H11 phosphorylation was

confirmed *in vitro* (Fig. 2D), and their influences on PGAM1 activity and pyruvate generation was validated as well (Fig. EV4A, B). **These observations suggested that PKM2 tetramerization blocks PGAM1 H11 phosphorylation, that is mediated by PKM2 dimer and monomer.**

Line 317-320, Page 12 (Discussion):

In the present study, it is also monomeric and dimeric PKM2 that were more efficient to phosphorylate and activate PGAM1 to augment glycolytic shunts, while the tetrameric PKM2 was not. This is quite reasonable because the tetramerization of PKM2 buries the appropriate interface that is accommodated for the substrate proteins' docking.

• In Figure 4, the Authors must show in a separate experiment that each kinase inhibitor is on target at the dose the Authors selected by showing that the drug abrogates phosphorylation of a target of the kinase.

Response: We thank the reviewer for this good suggestion. We treated A549 cells with Afatinib (EGFR inhibitor), Ruxolitinib (JAK inhibitor), Amuvatinib (c-Kit inhibitor) and Saracatinib (Src inhibitor), and then analyzed the phosphorylation levels of EGFR, STAT3, c-Kit and Src, which are regarded as the activation marks of EGFR, JAK, c-Kit and Src³. Our results demonstrated that the tested kinase inhibitors are indeed on target (new Fig. 4C). These interpretations and new data were integrated in the revised manuscript, shown as below:

Line 216-220, Page 9:

*To identify the tyrosine kinase(s) responsible for EGF-induced PGAM1 Y119 phosphorylation, EGFR and the tyrosine kinases downstream of EGFR were individually inactivated by inhibitors including Afatinib (EGFR inhibitor), Ruxolitinib (JAK family inhibitor), Amuvatinib (c-Kit inhibitor) or Saracatinib (Src family inhibitor). **The on-target effects of the inhibitors were validated³ (Fig. 4C).***

Fig. 4C:

- It is not clear why some western blots have quantitations and others do not. All western blots should be quantified from replicates, and statistical analysis performed, as is done in Figure 4.

Response: Following the reviewer's suggestion, we showed quantification with all western blots. Proper statistical analyses were provided as well.

- In Figure 5B, the metabolomics heat map is missing statistical analysis to show which of the metabolites were significantly different between cells with PGAM1 WT and PGAM1 Y119F.

Response: We thank the reviewer for this good suggestion. Per the reviewer's another concern about the detection of nucleotide synthesis, we carried out a new [¹³C₆]-glucose tracing experiment. A statistical analysis of the metabolites from different groups was shown in column configuration instead of the heat map (new Fig. 5B). For the convenience of the reviewer, it is shown as below:

Fig. 5B:

- On Results Section 5, the Authors state that "Because of the critical role of PPP and SSP in nucleotides synthesis and NADPH production, we further investigated the impact of PGAM1 Y119F mutation on these processes, and the result showed the impairment resulted from the mutation as expected (Fig. 5C and S7A)." However, Figure 5C is an Edu incorporation assay (which measures S phase in cell cycle), not a nucleotide synthesis assay. Is the nucleotide synthesis assay missing? Either include the missing assay and take out the Edu assay, or properly explain Figure 5c in the text as a proliferation assay.

Response: We fully agree with the reviewer that EdU incorporation is not a direct readout of nucleotide synthesis. To confirm the influence of PGAM1 Y119F mutation on nucleotide synthesis, we carried out new [¹³C₆]-glucose tracing experiment. The results revealed that along with the impairment of glycolysis shunts PPP and SSP, metabolic intermediates in nucleotide synthesis, such as ATP and AMP, were also significantly decreased (new Fig. 5B). This was consistent with the EdU incorporation assay that showed impaired DNA replication (Fig. 5C). The relevant statement and the new data were integrated in the revised manuscript, shown as below:

Line 248-252, Page 10:

Because of the critical role of PPP and SSP in nucleotides synthesis and NADPH production⁴⁻⁸, we further investigated the impact of PGAM1 Y119F mutation on these processes. The result showed that along with the impairment of glycolysis shunts PPP and SSP, metabolic intermediates in nucleotide synthesis, such as ATP and AMP, were also significantly decreased (Fig. 5B). This was consistent with the EdU incorporation assay that showed impaired DNA replication (Fig. 5C).

Fig. 5B, C:

• For Figure 6A: Since the PGAM1 pY119 antibody is a new antibody the Authors made, they need to show it is on-target and detecting Y119 in IHC by using a knockout or mutation model. The tumors from Figure 5F-G would work nicely this purpose.

Response: We thank the reviewer for this constructive suggestion. Using PGAM1 pY119 antibody, we tested Y119 phosphorylation of the tumor xenografts that expressed PGAM1^{WT} or PGAM1^{Y119F}, and the IHC data showed that Y119F mutation indeed diminished PGAM1 Y119 phosphorylation (new Fig. 5H). The relevant statement and the new data were integrated in the revised manuscript, and shown as below:

Line 262-264, Page 10:

In line with this observation, PGAM1 Y119 phosphorylation in Y119F mutant tumor xenografts was thoroughly wiped off (Fig. 5H); while Ki-67, a classic marker of cell proliferation, was also decreased due to PGAM1 Y119 mutation (Fig. 5H).

Fig. 5H:

• For Figure 6I, the Authors should show that the pY119-TAT peptide affects the glycolysis pathway in the tumors, for instance by looking at PGAM1 pY119, or some other direct marker.

Response: We appreciate the reviewer's suggestion that the impact from pY119-TAT peptide on glycolysis in the tumors should be evaluated. We re-performed the xenografts experiment. After the injection of the peptide, we analyzed H11 phosphorylation and the enzymatic activity of PGAM1 from the tumor xenografts. The results showed that pY119-TAT peptide significantly decreased H11 phosphorylation and PGAM1 enzymatic activity (new Fig. 7H, I), thereby slowing down the tumor growth. The relevant statement and the new data were integrated in the revised manuscript, and shown as below:

Line 293-297, Page 11:

Y119 phosphorylation, PKM2 binding, H11 phosphorylation and the enzymatic activity of PGAM1 from the tumor xenografts were further examined, and they were all significantly declined due to pY119-TAT peptide-injection (Fig. 7H, I). Collectively, pY119-containing peptide treatment interferes with the interaction of PGAM1-PKM2 and restricts tumor growth.

Fig. 7H, I:

Minor comments

• In the title for Results Section 3, the last word should be "recognition", "recognition" is not an English word.

Response: We have replaced “recognition” with “recognition” in the revised manuscript.

- In Results Section 3, "Intriguingly, PGAM1 is...", the correct word is "conserved", not "conversed".

Response: We corrected the spelling as suggested.

- In Figure 3F, the non-mammal species are not "lower" than mammals, this is scientifically incorrect. Please remove "lower" from the text and Figure.

Response: We modified the sentence as suggested.

- At the bottom of Results Section 3, the Authors state, "In addition, we also noticed that PGAM1^{WT} and PGAM1^{Y119F} no longer had different impact on PGAM1 H11 phosphorylation if PKM2 was depleted (Fig. 3I), further confirming that Y119-mutation-caused influence was inflicted by PKM2." I do not understand this sentence. I know the Authors do not conclude that PKM2 phosphorylates PGAM1 on Y119 since they address this in the next Figure, but this sentence, as written, suggests that. Please rewrite this sentence to more clearly state the conclusions.

Response: The previous statement was indeed obscure. We rephrased the sentence in the revised manuscript, which is shown as below:

Line 201-204, Page 8:

In addition, we also noticed that PGAM1^{WT} and PGAM1^{Y119F} no longer had different impact on PGAM1 H11 phosphorylation if PKM2 was depleted (Fig. 3I), which further confirmed that PGAM1 Y119 phosphorylation is prerequisite for PKM2-inflicted PGAM1 H11 phosphorylation.

Referee #2:

Increased expression/activation of the glycolytic enzyme phosphoglycerate mutase 1 (PGAM1) is a common event across a wide variety of cancers. A previous study from Vander Heiden et al. revealed that PGAM1 is activated by phosphorylation at the catalytic histidine residue (His11) with phosphoenolpyruvate serving as the phosphate donor (PMID: 20847263). The nature of the enzyme that catalyses phosphate transfer from PEP to PGAM1 was not identified by Vander Heiden and colleagues. In the current study, Wang et al. provide evidence that pyruvate kinase M2 (PKM2) acts as a histidine kinase to catalyse PGAM1 H11 phosphorylation. In addition, the authors demonstrate that phosphorylation of PGAM1 at Tyr119 is required for PKM2 binding and subsequent His11 phosphorylation. Finally, the authors provide evidence that a PGAM1-derived phospho-Tyr119 peptide or mutation of Tyr119 inhibits the interaction of PGAM1 with PKM2 to disrupt glycolytic metabolism and inhibit tumour growth. The conclusions in this paper are supported by robust and compelling experimental data. Given that this study reveals important insights into the regulation of glycolytic metabolism the findings should be of interest to researchers spanning multiple fields of research. My only comments/suggestions are minor and are outlined below.

Response: We appreciate the reviewer's recognition of the novelty and the significance of this work. Based on the constructive suggestions given by the reviewer, we carried out a number of new experiments. Hopefully, that would greatly improve the quality of the manuscript.

- With respect to Figure 4C/D, it would be good to include a readout of the activation state of the EGFR signalling pathway (e.g. ERK phosphorylation).

Response: We fully agree with the reviewer that it is important to show the activation of the EGFR signaling pathways for Figure 4 C/D. To this end, the phosphorylation levels of EGFR, STAT3, c-Kit and Src, in the presence or absence of the individual inhibitor were analyzed. The on-target effect of the particular inhibitor and transfection was validated³ (Fig. 4C, D). These interpretations and the new data were integrated in the revised manuscript and highlighted, shown as below:

Line 216-220, Page 9:

To identify the tyrosine kinase(s) responsible for EGF-induced PGAM1 Y119 phosphorylation, EGFR and the tyrosine kinases downstream of EGFR were individually inactivated by inhibitors including Afatinib (EGFR inhibitor), Ruxolitinib (JAK family inhibitor), Amuvatinib (c-Kit inhibitor) or Saracatinib (Src family inhibitor). The on-target effects of the inhibitors were validated³ (Fig. 4C).

Line 230-233, Page 9:

Additionally, overexpression of Src enhanced PGAM1 Y119 phosphorylation, PGAM1-PKM2 interaction as well as PGAM1 H11 phosphorylation in A549 cells that expressed wild type PGAM1 but not PGAM1 Y119F (Fig. 4D), suggesting the commitment of Src in transmission of growth factors' signaling to induce PGAM1 Y119 phosphorylation.

Fig. 4C, D:

- On page 10, the authors state "NADPH is not only a classic antioxidant that quenches ROS...". NADPH is not antioxidant. Instead, NADPH provides reducing equivalents to generate reduced forms of antioxidant molecules. This statement should be corrected.

Response: We thank the reviewer for this reminder, and we corrected the statement about NADPH in the revised manuscript, shown as below:

Line 254, Page 10:

*NADPH is not only producing reduced forms of antioxidants, but also involved in lipid synthesis*⁹.

- In Figure 5F, WT A549 xenografts grew to an average size of approximately 600 mm³ after 14 days. In Figure 6I, WT A549 xenografts grew to only approximately 200 mm³ after 14 days. While differences in tumour growth kinetics between experiments are not unusual, why were the experiments in Figure 6I stopped when tumours were so small?

Response: As the reviewer pointed out, it is not unusual that the tumour growth kinetics display differences between experiments. To further confirm the function of pY119-TAT, we set up new animal experiments, and when the tumor volumes were approximately up to 500 mm³, the experiments stopped (new Fig. 7D-F). Injection of pY119-TAT significantly slowed down the growth of tumour xenografts (new Fig. 7D-F). For the convenience of the reviewer, the re-written interpretation and the new data are shown as below:

Line 286-292, Page 11:

To evaluate the therapeutic potential of pY119-TAT peptide *in vivo*, A549 cells were subcutaneously injected into athymic nude mice (Fig. 7C). Five days after tumor colonization, 10 mg/kg of TAT or pY119-TAT peptide was injected intraperitoneally every other day. Another ten days after peptide injection, when the tumor xenografts were up to 500 mm³ approximately in the TAT peptide-

administrated mice, samples were harvested. Both tumor volume and mass in the mice injected with pY119-TAT peptide were significantly less compared with that injected with TAT peptide (Fig. 7D-F).

Fig. 7D-F

- With respect to Figure 6I, animal bodyweight over the course of the experimental period should be included as supplementary data.

Response: Following the reviewer's suggestion, we have performed new xenograft experiments and included the animal bodyweight data in the revised manuscript. We found that pY119-TAT peptide injection had no effect on the mice's bodyweight (new Fig. 7G). For the convenience of the reviewer, it is shown as below:

Line 291-293, Page 11:

Both tumor volume and mass in the mice injected with pY119-TAT peptide were significantly less compared with that injected with TAT peptide (Fig. 7D-F); whereas, the mice's bodyweight had no significant difference between the two groups (Fig. 7G).

Fig. 7G:

- With respect to Figure 6I, pY119-TAT peptide was administered from the time of tumour cell injection. It would be more impactful to demonstrate the impact of pY119-TAT peptide on tumour growth once tumours are already established.

Response: We agree with the reviewer that It would be more impactful to demonstrate the

impact of pY119-TAT peptide on tumour growth once tumours are already established. We set up new animal experiments. The pY119-TAT peptide was injected to the mice after A549 cells were colonized for 5 days, when tumour is established, instead of being dosed immediately after the subcutaneous inoculation of A549 cells (new Fig. 7C). Administration of the peptide significantly slowed down the tumour growth (new Fig. 7D-I). For the convenience of the reviewer, the re-written interpretations and the new data are shown as below:

Line 286-297, Page 11:

To evaluate the therapeutic potential of pY119-TAT peptide in vivo, A549 cells were subcutaneously injected into athymic nude mice (Fig. 7C). Five days after tumor colonization, 10 mg/kg of TAT or pY119-TAT peptide was injected intraperitoneally every other day. Another ten days after peptide injection, when the tumor xenografts were up to 500 mm³ approximately in the TAT peptide-administrated mice, samples were harvested. Both tumor volume and mass in the mice injected with pY119-TAT peptide were significantly less compared with that injected with TAT peptide (Fig. 7D-F); whereas, the mice's bodyweight had no significant difference between the two groups (Fig. 7G). Y119 phosphorylation, PKM2 binding, H11 phosphorylation and the enzymatic activity of PGAM1 from the tumor xenografts were further examined, and they were all significantly declined due to pY119-TAT peptide-injection (Fig. 7H, I). Collectively, pY119-containing peptide treatment interferes with the interaction of PGAM1-PKM2 and restricts tumor growth.

Fig. 7C-I

- The manuscript should be carefully edited to correct grammatical errors, spelling mistakes and issues with sentence structure.

Response: We apologize for such mistakes. We carefully corrected the grammatical errors, spelling mistakes and the issues with sentence structure.

Referee #3:

The manuscript by Wang et al reported identification of PKM2 as a "moonshine" histidine kinase in a phosphoenolpyruvate (PEP)-dependent manner to catalyze PGAM1 H11 phosphorylation. The authors also showed that stimulation by EGF activates Src, which in turn phosphorylates PGAM1 at Y119. The Y119 phosphorylation contributes to recruitment of PKM2 to PAGM1 and achieving H11 phosphorylation. Lastly, the authors claimed that disruption of PGMA1-PKM2 binding by a PGAM1-derived pY119-containing cell permeable peptide or Y119 mutation on PGAM1 reduces H11 phosphorylation of PGAM1, leading to reduced tumor growth.

Overall the study is well conducted and well controlled, and the findings are of some interests with mechanistic insights. This reviewer only has a couple of concerns that need to be addressed by additional experimental evidence:

Response: We appreciate the reviewer's recognition of the quality of the study as well as the insight into the molecular mechanism we proposed. We also thank the reviewer for his/her concerns. We did a number of experiments to address these concerns. Hopefully, the additional evidence would substantially convince the mechanism we proposed and greatly improve the impact of the present work.

- 1. It is well known that in mammalian cells, the cofactor 2, 3-BPG of PGAM1 is responsible for phosphorylation of H11 by transferring its phosphate group at C3 position to H11. Matt Vander Heiden reported in 2010 that 2,3-BPG levels were increased in PKM2-expressed cells as compared to PKM1-expressed cells (Vander Heiden, 2010, Science, 329, 1492-1499). Thus, it is possible that in cells, PKM2 may increase 2,3-BPG levels for phosphorylation of PGAM1 at H11. The authors need to convincingly exclude this possibility before draw the conclusion that PKM2 phosphorylates PGAM1 directly and using PEP as the phosphate providing, because all of the studies performed are in vitro that cannot fully represent the physiological situation.

Response: The reviewer's concern is quite relevant; however, it is somehow also challenging our proposed mechanism. We are pleased to address this concern of the reviewer because the additional evidence would substantially convince the mechanism we proposed and greatly improve the impact of the present work.

The 2,3-bisphosphoglycerate (2,3-BPG), an intermediate in PGAM1-catalyzed conversion of 3-PG into 2-PG, is responsible for phosphorylation of H11 by transferring its phosphate group at C3 position to H11. This raises the possibility that 2,3-BPG may contribute to PGAM1 H11 phosphorylation independent of PEP-based phosphate group donation because 2,3-BPG level is increased in PKM2-expressed cells as reported¹.

Based on the previous studies, the flexible residues in PGAM1, such as Y92, R116 or R117, were shown to be important for cofactor 2,3-BPG binding¹⁰⁻¹². Another study also reported that Y26 phosphorylation could improve 2,3-BPG binding and PGAM1 H11 phosphorylation through releasing the inhibition of E19¹³. To exclude the involvement of 2,3-

BPG in PKM2-mediated PGAM1 phosphorylation in the present study, we constructed PGAM1 mutants including Y26F and Y92F, and performed drug-affinity responsive target stability (DARTS) assays and isothermal titration calorimetry (ITC) measurements, the techniques used to determine the binding of small molecules to macromolecules. The data verified that Y26F and Y92F mutations indeed inhibited the interaction between PGAM1 and 2,3-BPG (new Fig. 1K, L). Thereafter, PKM2 were overexpressed along with WT PGAM1 or its mutations in A549 cells, the level of PGAM1 H11 phosphorylation was analyzed. Although compared with the wild type PGAM1, Y26F and Y92F mutations significantly decreased H11 phosphorylation, that was resulted from the disruption of 2,3-BPG binding with PGAM1 to provide phosphate group, the interactions of the mutants with PKM2 were not impaired, and the overexpression of PKM2 could still significantly increase H11 phosphorylation of the PGAM1's Y26F and Y92F mutants (new Fig. 1M). Taken together, data suggested that PKM2-catalyzed PGAM1 H11 phosphorylation is independent of 2,3-BPG. The interpretation of the results and the corresponding data were integrated in the revised manuscript, and for the convenience of the reviewer, shown as below:

Line 122-144, Page 5-6:

2,3-Bisphosphoglycerate (2,3-BPG), an intermediate in PGAM1-catalyzed conversion of 3-PG into 2-PG, is responsible for phosphorylation of H11 by transferring its phosphate group at C3 position to H11. This raises the possibility that 2,3-BPG may contribute to PGAM1 H11 phosphorylation independent of PEP-derived phosphorus group donation since 2,3-BPG level is increased in PKM2-expressed cells as reported^{1, 14}. To exclude the involvement of 2,3-BPG in the PKM2-catalyzed PGAM1 H11 phosphorylation. PGAM1 mutants, such as PGAM1Y26F and PGAM1Y92F were constructed to disrupt 2,3-BPG binding as described in the previous studies^{10, 12, 13}. The isothermal titration calorimetry (ITC) measurement was employed, and the result verified that Y26F and Y92F mutations thoroughly inhibited the interaction between the recombinant PGAM1 and 2,3-BPG in vitro (Fig. 1K). In addition, the drug affinity responsive target stability (DARTS) assay, that is used to detect the reduction in the protease susceptibility of the target proteins upon the small molecules binding, was conducted¹⁵. A549 cell lysates containing WT PGAM1 or its mutants were incubated with increasing concentration of 2,3-BPG, then digested with pronase. The result showed that, 2,3-BPG was able to protect the WT PGAM1, but not that with Y26F or Y92F mutation from the degradation by pronase, implying a failure of 2,3-BPG binding with the mutants (Fig. 1L). Next, PKM2 were overexpressed along with WT PGAM1 or its mutants in A549 cells, and the PGAM1 H11 phosphorylation in the presence or absence of PKM2 was detected. Although compared with the WT PGAM1, H11 phosphorylation of Y26F and Y92F mutants significantly decreased, that was resulted from the disruption of 2,3-BPG binding with PGAM1 to provide phosphorus group, the interactions of the mutants with PKM2 were not impaired, and the overexpression of PKM2 could still significantly increase H11 phosphorylation of PGAM1Y26F and PGAM1Y92F mutants (Fig. 1M), suggesting that 2,3-BPG is not involved in PKM2-catalyzed PGAM1 H11 phosphorylation. Taken together, combined data supported that PKM2 acts as a histidine kinase to phosphorylate PGAM1 at H11 in a PEP-dependent manner.

Fig. 1K-M

- 2. The authors need to convincingly demonstrate whether endogenous PKM2 and PGAM1 interact in diverse tumor cells, and whether such binding is regulated by EGF stimulation. Moreover, PGAM1 was reported to be phosphorylated at Y119 in cells expressing oncogenic FGFR1 (Hitosugi, 2013, Nature Communications). Is it all depending on Src? Do diverse oncogenic tyrosine kinases including FGFR1, FGFR3, ALK, PDGFRA, PDGFRB etc directly phosphorylate PGAM1 at Y119 or they have to activate Src for Y119 phosphorylation?

Response: Following the reviewer's suggestion, we confirmed that endogenous PGAM1 interacted with PKM2 in diverse tumor cells including A549, H1299, MCF-7, MDA-MB-231, AGS and Jurkat (Fig. 1F and EV2A-C). Moreover, EGF stimulation significantly enhanced endogenous PGAM1-PKM2 interaction in A549 and MCF-7 cells (new Fig. 4A and EV7A). For the convenience of the reviewer, the statement and corresponding data are shown as below:

Line 92-96, Page 4-5:

PKM2-PGAM1 interaction was further confirmed in a panel of highly proliferating tumor cells, including A549, H1299, MCF-7, MDA-MB-231, AGS and Jurkat, but hardly detectable in untransformed cells, including HBE-2, MCF-10A, GES-1, activated T and naïve T cells (Fig. 1F and EV2A-C), suggesting a pan-cancer existence of the interaction between these two metabolic enzymes.

Fig. 1F and EV2A-C

Line 212-215, Page 8-9:

Herein, we further explored the role of EGF signaling in the regulation of PGAM1 by PKM2. In response to EGF signaling, increases in PGAM1 Y119 phosphorylation, the interaction of PKM2 with PGAM1, as well as PGAM1 H11 phosphorylation were observed (Fig. 4A, B and EV7A, B)

Fig. 4A and EV7A

Also, it is interesting to test whether different oncogenic tyrosine kinases could phosphorylate PGAM1 at Y119 directly. Our new experiments showed that FGF and PDGF stimulation promoted Y119 phosphorylation and PGAM1-PKM2 interaction, and then treatment of inhibitors targeting FGFR, PDGFR or Src blocked the effects of FGF or PDGF (new Fig. EV7C, D). Moreover, we carried out in vitro kinase assay and found that only Src, but not EGFR, FGFR, PDGFR, directly phosphorylated PGAM1 at Y119 (new Fig. 4E). So, PGAM1 Y119 phosphorylation was triggered by Src, downstream of diverse oncogenic receptor and non-receptor tyrosine kinases, might including FGFR1, FGFR3, PDGFRA,

PDGFRB, ALK etc. For the convenience of the reviewer, the interpretation of the results and new figures are shown as below:

Line 224-239, Page 9:

Besides EGF, other oncogenic growth factors, such as fibroblast growth factor (FGF) and platelet-derived growth factor (PDGF), could activate Src and widely function in various tumors¹⁶⁻¹⁹. Thus, the effects of different oncogenic growth factors on induction of PGAM1 Y119 phosphorylation were further tested. FGF or PDGF stimulation promoted Y119 phosphorylation and PGAM1-PKM2 interaction, and the inhibition of FGF receptor (FGFR), PDGF receptor (PDGFR) or Src individually by Zoligratinib, Crenolanib or Saracatinib blocked the functions of FGF or PDGF (Fig. EV7C, D). Additionally, overexpression of Src enhanced PGAM1 Y119 phosphorylation, PGAM1-PKM2 interaction as well as PGAM1 H11 phosphorylation in A549 cells that expressed wild type PGAM1 but not PGAM1 Y119F (Fig. 4D), suggesting the commitment of Src in transmission of growth factors' signaling to induce PGAM1 Y119 phosphorylation. To further validate the function of Src, in vitro kinase assay was carried out and the result showed that only Src, but not EGFR, FGFR, PDGFR, directly phosphorylated PGAM1 at Y119 (Fig. 4E, F); however, Y119F mutation is unable to respond to Src-mediated-Y119 phosphorylation, thus failing to regulate H11 phosphorylation (Fig. 4F). Taken together, these data confirmed the function of Src in growth factors' signaling-induced PGAM1 Y119 phosphorylation, which in turn enhances PKM2-PGAM1 interaction as well as PGAM1 H11 phosphorylation.

Fig. EV7C, D and 4D, E:

1. Vander Heiden, M.G. *et al.* Evidence for an alternative glycolytic pathway in rapidly proliferating cells. *Science* **329**, 1492-1499 (2010).
2. Wang, P., Sun, C., Zhu, T. & Xu, Y. Structural insight into mechanisms for dynamic regulation of PKM2. *Protein Cell* **6**, 275-287 (2015).
3. Liu, R. *et al.* Tyrosine phosphorylation activates 6-phosphogluconate dehydrogenase and promotes tumor growth and radiation resistance. *Nat Commun* **10**, 991 (2019).
4. Stincone, A. *et al.* The return of metabolism: biochemistry and physiology of the pentose phosphate pathway. *Biol Rev Camb Philos Soc* (2014).
5. Patra, K.C. & Hay, N. The pentose phosphate pathway and cancer. *Trends Biochem Sci* **39**, 347-354 (2014).
6. Jiang, P., Du, W.J. & Wu, M.A. Regulation of the pentose phosphate pathway in cancer. *Protein Cell* **5**, 592-602 (2014).
7. Fan, J. *et al.* Quantitative flux analysis reveals folate-dependent NADPH production. *Nature* **510**, 298-302 (2014).
8. Hitosugi, T. *et al.* Phosphoglycerate mutase 1 coordinates glycolysis and biosynthesis to promote tumor growth. *Cancer Cell* **22**, 585-600 (2012).
9. Fan, J. *et al.* Quantitative flux analysis reveals folate-dependent NADPH production. *Nature* **510**, 298-302 (2014).
10. Wang, Y.L. *et al.* Seeing the process of histidine phosphorylation in human bisphosphoglycerate mutase. *J Biol Chem* **281**, 39642-39648 (2006).
11. Bond, C.S., White, M.F. & Hunter, W.N. Mechanistic implications for cofactor-dependent phosphoglycerate mutase based on the high-resolution crystal

- structure of a vanadate complex. *J Mol Biol* **316**, 1071-1081 (2002).
12. Wang, Y.L. *et al.* Crystal structure of human B-type phosphoglycerate mutase bound with citrate. *Biochem Biophys Res Commun* **331**, 1207-1215 (2005).
 13. Hitosugi, T. *et al.* Tyr26 phosphorylation of PGAM1 provides a metabolic advantage to tumours by stabilizing the active conformation. *Nat Commun* **4**, 1790 (2013).
 14. Wiese, E.K. & Hitosugi, T. Tyrosine Kinase Signaling in Cancer Metabolism: PKM2 Paradox in the Warburg Effect. *Front Cell Dev Biol* **6** (2018).
 15. Lomenick, B. *et al.* Target identification using drug affinity responsive target stability (DARTS). *Proc Natl Acad Sci USA* **106**, 21984-21989 (2009).
 16. Ostman, A. *et al.* PDGF receptors as cancer drug targets. *Ann Oncol* **16**, 21-21 (2005).
 17. Cao, Y.H. Multifarious functions of PDGFs and PDGFRs in tumor growth and metastasis. *Trends Mol Med* **19**, 460-473 (2013).
 18. Presta, M., Chiodelli, P., Giacomini, A., Rusnati, M. & Ronca, R. Fibroblast growth factors (FGFs) in cancer: FGF traps as a new therapeutic approach. *Pharmacol Therapeut* **179**, 171-187 (2017).
 19. Turner, N. & Grose, R. Fibroblast growth factor signalling: from development to cancer. *Nat Rev Cancer* **10**, 116-129 (2010).

Dear Dr Wei,

Thank you for submitting your revised manuscript (EMBOJ-2023-115733R) to The EMBO Journal. Your amended study was sent back to the three referees for their scientific re-evaluation, and we have received detailed comments from all of them, which I enclose below. As you will see, the experts state that the work has been substantially improved by the revisions and they are now broadly in favour of publication.

Thus, we are pleased to inform you that your manuscript has been accepted in principle for publication in The EMBO Journal.

We now need you to take care of a number of issues related to formatting and data presentation as detailed below, which should be addressed at re-submission.

Please contact me at any time if you have additional questions related to below points.

As you might have seen on our web page, every paper at the EMBO Journal now includes a 'Synopsis', displayed on the html and freely accessible to all readers. The synopsis includes a 'model' figure as well as 2-5 one-short-sentence bullet points that summarize the article. I would appreciate if you could provide this figure and the bullet points.

Thank you for giving us the chance to consider your manuscript for The EMBO Journal. I look forward to your final revision.

Again, please contact me at any time if you need any help or have further questions.

Kind regards,

Daniel Klimmeck

>> Adjust the title of the 'Conflict of Interest' section to 'Disclosure and Competing Interests Statement' and move after Acknowledgements.

>> Funding: please enter 'the Young Scientific and Technological Talents Support Project of Jilin Province (QT202105)' into our online manuscript system. Merge with Acknowledgements in the manuscript.

>> Reference format: needs a heading added, change from numerical to alphabetical order, 10 author names should be listed before et al. .

>> Introduce ORCID IDs for all corresponding authors (X. B.) via our online manuscript system. Please see below for additional information.

>> Appendix: there are currently 10 EV figures. Reduce to max five; the other supplemental figures should be compiled in a PDF labelled "Appendix", with a ToC and the figure legends included, and with the nomenclature "Appendix Figure S1" etc. . Adjust callouts in the manuscript text and legends accordingly.

>> Data availability section: Please remove the referee tokens and make sure data privacy is released.

>> Please cite referencing of your previous work (PMID: 35228743) in the Material and Methods section.

>> Consider additional changes and comments from our production team as indicated below:

- DAS:

1. Please note that the specific URLs for PXD049310 and PXD049235 datasets are not provided in the data availability statement.
 2. Please note that reviewer access codes for PXD049310 and PXD049235 datasets are provided in the manuscript.
- Figure legends:
1. Please note that a separate 'Data Information' entitled section is required in the legends of figures 1a, d, f-j, l, m; 2a, c-d; 3a, c-f, h-i; 4a-f; 7b, e-i; EV 1a-e; EV 2a-e; EV 3a-h; EV 7a-d; EV 8a-e; EV 7a-b, d.
 2. Please note that the legends for figures EV 1c-e is not provided in the sequential manner (legend for figure EV 1e is provided before legend of figures EV 1c-d). This needs to be rectified.
 3. Please note that the legend for figure EV 3c is missing in the manuscript. This needs to be rectified.
 4. Please indicate the statistical test used for data analysis in the legends of figures 2b; 6b-c.
 5. Please note that in figures 2a, c-d; 7b, e-i; there is a mismatch between the annotated p values in the figure legend and the annotated p values in the figure file that should be corrected.
 6. Please define the annotated p values ***/** in the legend of figure 6b-c; as appropriate.
 7. Please note that information related to n is missing in the legend of figure 6b.
 8. Although 'n' is provided, please describe the nature of entity for 'n' in the legends of figures 5a-b; 6e; 7b, h-i; EV 1a-e; EV 2a-e; EV 4a-b; EV 7a-d; EV 8a-e; EV 9a-b, d.

Referee #1:

I thank the Authors for very carefully responding to all my prior comments with new experiments and appropriate text changes. I also find the responses to the other Reviewers to be reasonable and well-justified. I have no further comments or concerns.

Referee #2:

The authors have addressed my comments and I am satisfied with the revised manuscript. This is a robust study that reveals important new mechanistic insights regarding the regulation of cellular metabolism.

Referee #3:

The authors have satisfactorily addressed this reviewer's concerns and the revised manuscript is very much strengthened.

The authors addressed the minor editorial issues.

Dear Dr Wei,

Thank you for submitting the revised version of your manuscript. I have now evaluated your amended manuscript and concluded that the remaining minor concerns have been sufficiently addressed.

I am pleased to inform you that your manuscript has been accepted for publication in the EMBO Journal.

On a different note, I would like to alert you that EMBO Press offers a format for a video-synopsis of work published with us, which essentially is a short, author-generated film explaining the core findings in hand drawings, and, as we believe, can be very useful to increase visibility of the work. Please see the following link for representative examples and their integration into the article web page:

<https://www.embopress.org/doi/full/10.15252/emboj.2019103932>

Kind regards,

Daniel Klimmeck

Daniel Klimmeck, PhD
Senior Editor
The EMBO Journal
EMBO
Postfach 1022-40
Meyerhofstrasse 1
D-69117 Heidelberg
contact@embojournal.org
Submit at: <http://emboj.msubmit.net>
